# Asymmetric Dual Self-Distillation for 3D Self-Supervised Representation Learning

**Remco F. Leijenaar**\*    **Hamidreza Kasaei**
Department of AI, University of Groningen, The Netherlands

## Abstract

Learning semantically meaningful representations from unstructured 3D point clouds remains a central challenge in computer vision, especially in the absence of large-scale labeled datasets. While masked point modeling (MPM) is widely used in self-supervised 3D learning, its reconstruction-based objective can limit its ability to capture high-level semantics. We propose AsymDSD, an Asymmetric Dual Self-Distillation framework that unifies masked modeling and invariance learning through prediction in the latent space rather than the input space. AsymDSD builds on a joint embedding architecture and introduces several key design choices: an efficient asymmetric setup, disabling attention between masked queries to prevent shape leakage, multi-mask sampling, and a point cloud adaptation of multi-crop. AsymDSD achieves state-of-the-art results on ScanObjectNN (90.53%) and further improves to 93.72% when pretrained on 930k shapes, surpassing prior methods.

## 1   Introduction

As domains such as robotics, autonomous driving, AR/VR, and remote sensing continue to grow, the importance of three-dimensional (3D) data becomes increasingly pronounced. A central challenge in 3D vision lies in learning semantically meaningful representations from unstructured 3D point clouds. In contrast to 2D computer vision—where large labeled datasets like ImageNet [1] have played a prominent role in driving progress—3D datasets remain limited in both scale and diversity (see Fig. 1). This scarcity, which has been referred to as the *data desert* problem [2], is exacerbated by the difficulty of annotating 3D data, particularly at the point level [3]. Although recent efforts such as Objaverse [4, 5] mark progress toward web-scale 3D data collection, they still lag behind the scale achieved in 2D vision and natural language processing (NLP). The lack of labeled 3D data has fueled a growing interest in self-supervised representation learning (SSL/SSRL) for 3D understanding [6]. SSL has proven highly effective in both NLP [7, 8] and 2D vision [9–12], offering strong scalability [13–16], and robust down-stream capabilities [17, 18, 16], even with minimal labeled data [19]. These successes have inspired SSRL techniques to be adapted to point cloud data. Particularly, masked point modeling (MPM) approaches have gained traction [20–26], given their effectiveness and conceptual alignment with masked modeling frameworks for 2D and NLP.

Despite their popularity, we argue that MPM-based approaches have fundamental limitations. Reconstruction objectives tend to emphasize short-range dependencies and high-frequency details [27, 28], which are often dominated by noise rather than semantically meaningful structure. This issue is exacerbated in complex 3D geometries, where undersampling introduces high target variance. More broadly, MPM may not lead to the semantic abstraction crucial for robust downstream performance. This is backed by empirical findings that demonstrate that these models underperform compared to invariance-based alternatives in low-shot and linear probing regimes [19]. To overcome these limitations, several works have attempted to fuse generative reconstruction with invariance-based objectives [21, 26]. However, the pattern differences of these objectives can cause them to interfere

---

\*Code available at `https://github.com/RFLeijenaar/AsymDSD`

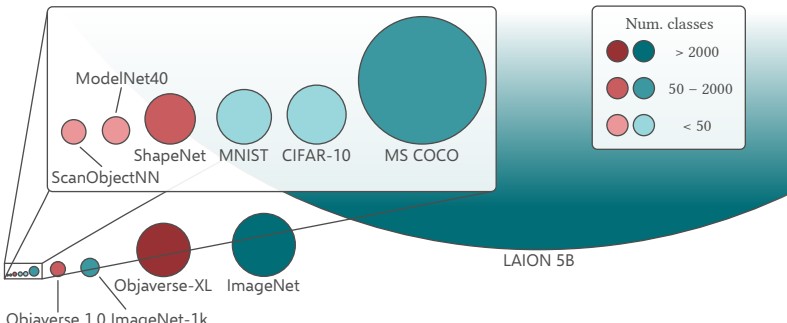

Figure 1: A depiction of the size of some well-known predominantly object-centered datasets in the computer vision domain. The blue color corresponds to (2D) image datasets, and red to 3D datasets.

with one another, resulting in worse performance than using reconstruction alone. Notably, ReCon [26] solves this through a two-tower architecture that separates the objectives, but this comes with significantly increased computational overhead.

In light of these issues, we seek a more elegant integration of local masked modeling and global invariance learning. Particularly, we posit a reframing of the MPM paradigm: rather than predicting the input from the latent, what if we instead predict the latent representations themselves? This shift aligns with the philosophy behind works such as CPC [29], data2vec [30, 31], and I-JEPA [32], that argue for learning semantics through prediction in latent space. The intuition stems from the notion that semantically meaningful features are those that are predictive across spatial contexts, even when fine details are occluded. For example, given only a visible wing of an airplane, one cannot reasonably predict the exact geometry of the tail, but one can predict the presence of a tail—a semantic category.

Building on this insight, we introduce **AsymDSD**: an **Asym**metric **D**ual **S**elf-**D**istillation framework for 3D point clouds (Fig. 2). AsymDSD effectively combines predictive masked modeling with invariance learning, using a joint embedding architecture (JEA) trained through self-distillation from a momentum teacher. The framework is designed with efficiency in mind, and addresses issues such as representation collapse and shape leakage, taking inspiration from a variety of recent works in SSRL on images. We summarize the core components and main contributions as follows:

- **Dual Self-Distillation Objectives**: The model jointly optimizes (1) a patch-level latent masked point modeling (MPM) objective through same-view self-distillation on masked tokens, and (2) a global invariance learning objective using cross-view self-distillation [33, 18]. The representations are projected to a distribution over a discrete latent variable, thereby explicitly modeling a posterior. This gives more direct control to overcome representation collapse and stabilize training.

- **Asymmetric Architecture**: The student model, unlike the teacher, hosts an encoder-predictor design. Given effective high mask ratios [22], the relatively heavy encoder processes only a small number of visible patches, while a lightweight predictor builds the representations of masked patches. Unlike previous methods [34, 22], we disable self-attention over the mask queries, which avoids leaking the global shape through the positional queries, and further enhances efficiency.

- **Multi-Mask**: To amortize the cost of the additional teacher, we introduce *multi-mask* [31], which samples multiple independent masks per point cloud. This allows targets and non-contextualized student embeddings to be reused across masks, effectively increasing batch size at minimal cost.

- **Multi-Crop**: While many point cloud-specific augmentations fall short in enforcing a challenging invariance objective, the modality-agnostic cropping augmentations proves effective. In particular, we adapt *multi-crop* [35] to point clouds, encouraging the model to efficiently learn robust local-to-global mapping capabilities.

To evaluate the effectiveness of AsymDSD, we adopt the common ShapeNet [36] pretraining protocol and transformer backbone [21], ensuring a fair comparison. Under this setup, AsymDSD achieves state-of-the-art performance on ScanObjectNN [37], reaching 90.53% accuracy, which is a 5.35% absolute improvement over the Point-MAE baseline [22]. The method also exhibits strong generalization in few-shot settings, as demonstrated on ModelNet40 [38]. Furthermore, we run ablations on our training framework to show that the proposed components contribute substantial improvements.

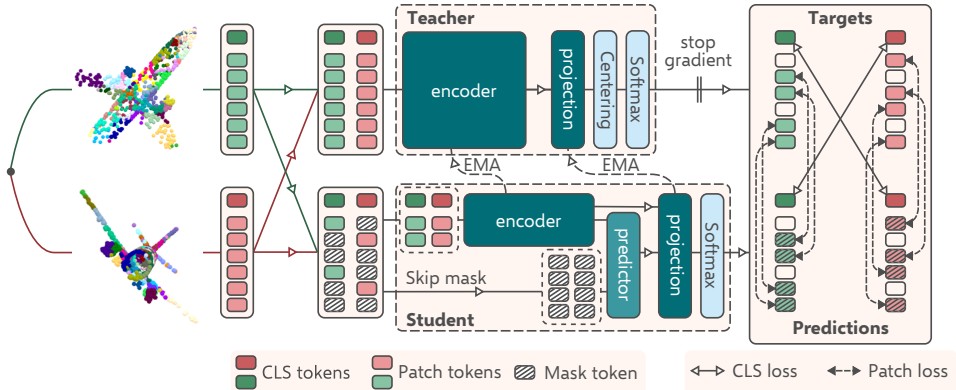

Figure 2: High level overview of **AsymDSD**. The diagram highlights the asymmetry between the teacher and student networks, and shows the distillation of knowledge from the momentum encoded (EMA) teacher on both a cross-view global (CLS) and same-view patch level. The block widths indicate the number of patches processed, while their lengths represent network depth. The student's efficient design is reflected in its deep but narrow encoder and wide but shallow predictor.

To assess scalability, we pretrain AsymDSD on a large composite dataset incorporating synthetic and scanned point clouds, including Objaverse [4], totaling over 930,000 shapes. Pre-training on this large dataset attains 93.72% accuracy on ScanObjectNN, a new single-modal SOTA with a standard transformer, exceeding PointGPT-L [24] by 2.6%.

## 2 Related Work

**SSRL in 2D Vision** has evolved along two main paradigms: *discriminative* (or invariance-based) and *generative* approaches. Discriminative methods primarily focus on overcoming *representation collapse*: a mode wherein representations become constant, and thus independent of the input. Approaches such as SimCLR [10] and MoCo [39] employed contrastive learning, which relies on comparing positive pairs (augmentations of the same image) against a large set of negatives. However, these methods require large batch sizes or memory banks to be effective. Subsequent works, such as BYOL [9] and SimSiam [40], demonstrated that contrastive negatives are dispensable. They utilize a student-teacher setup with architectural asymmetry to stabilize training and avoid collapse. Follow-up studies further explored this space by integrating regularization techniques like mean entropy maximization [41, 19], clustering constraints [35], and representation decorrelation [42, 43].

Parallel to this, denoising autoencoders [44], have evolved into masked image modeling (MIM) approaches [45, 12]. MAE, in particular, employs an efficient encoder-decoder setup that leverages the transformer's capacity to process sparse inputs. BEiT [46] introduced discrete token prediction using a pre-trained dVAE [47], however its targets lack high-level semantics [33]. Hybrid methods have since emerged that blend MIM with joint embedding architectures. Notably, data2vec [30, 31] and I-JEPA [32] reframe MIM as latent representation prediction, regressing contextualized embeddings produced by a momentum teacher instead of raw pixels. Differently, iBOT [33] and DINOv2 [18] include a global invariance objective to ensure semantically rich targets for the MIM objective.

**SSRL for Point Clouds.** The success of SSRL in 2D vision has inspired analogous developments in 3D point cloud learning. Early invariance-based methods adopted contrastive frameworks [48–50] but also BYOL-style [51] or DINO-style [52] training. With the adoption of the standard transformer, point cloud-specific adaptations of MIM emerged. Point-BERT [21] and Point-MAE [22] employ masked token prediction strategies analogous to their image counterparts. To avoid shape leakage through the mask queries, PointGPT [24] sequentializes the point patches to enable autoregressive modeling. In contrast, we demonstrate that a simpler and more efficient alternative—disabling attention on mask tokens—is equally effective. point2vec [34] builds on data2vec [30] with an added predictor module, yet it does not address shape leakage at the predictor stage. ReCon [26] successfully combines masked point modeling (MPM) with multi-modal invariance learning but introduces added complexity through its dual-tower architecture. In comparison, our method seamlessly incorporates invariance learning without any changes to the underlying model architecture.

# 3 AsymDSD: Asymmetric Dual Self-Distillation

AsymDSD unfies two complementary self-supervised objectives, which we initially introduce as independent SSRL approaches. While not tied to a specific model architecture, we present it under a ViT-style [53] backbone with a lightweight PointNet-based patch embedding module, following Point-BERT [21]. A more detailed description of this architecture can be found in Appendix A.

## 3.1 Global Objective: Invariance learning

The global objective of AsymDSD innvolves the learning of representations with an abstraction of semantics that is discriminative to individual inputs, up to a set of data augmentations. For this purpose, it integrates knowledge distillation [54] and invariance learning in an end-to-end framework. Specifically, given an input sample $x \sim p(x)$, latent variable $z \in \mathcal{Z}$, and two augmentations $t_1, t_2 \sim \mathcal{T}$, the invariance-based distillation objective is:

$$\theta^* = \underset{\theta \in \Theta}{\operatorname{argmin}} \, \mathbb{E}_{x \sim p(x), \, t_1, t_2 \sim \mathcal{T}} \left[ D_{\mathrm{KL}} \left( p_{\theta'}^t(z \mid t_1(x)) \, \| \, p_\theta^s(z \mid t_2(x)) \right) \right] \tag{1}$$

where $p_\theta(z \mid x) = \operatorname{softmax}(f_\theta(x)/\tau)_z$, with $\tau$ a temperature to control the sharpness. In SSRL, the teacher posterior $p_{\theta'}^t(z|x)$ is not known ahead of training. Instead, past iterations of the student serve as a proxy for the teacher. Particularly, we use an exponential moving average (EMA) of the student's parameter: $\theta' \leftarrow \eta\theta' + (1 - \eta)\theta$, where $\theta'$ are the teacher and $\theta$ the student parameters, and $\eta$ a decay rate.

Iteratively optimizing Equation 1, may lead to pathological solutions in absence of additional constraints. A 'shortcut' in minimizing any such objective is for the latent $z$ to become independent of the input $x$, i.e. $p_\theta(z) = p_\theta(z \mid x)$. This is particularly implied by the scenarios:

1. $H\left(p_\theta(z)\right) = H(\mathbb{E}_{x \sim p(x)}[p_\theta(z \mid x)]) = 0$, i.e., the marginal entropy becomes zero. The model collapses to a single latent representation $z$ that is always assigned a probability of 1.

2. $H(p_\theta(z \mid x)) = \log |\mathcal{Z}|$, i.e., the posterior entropy becomes maximal. The model always outputs a uniform distribution over the latent space $\mathcal{Z}$.

To overcome these modes of posterior collapse, we follow DINO [11], by applying *centering* and *sharpening*. Centering prevents scenario (1) by subtracting a running mean from the teacher logits, thereby reducing the logits of the latent $z$ that are frequently assigned high probability, thus helping to avoid low-entropy collapse. Sharpening prevents scenario (2) by setting a lower teacher softmax temperature compared to the student: $\tau_t < \tau_s$. This makes the posterior distribution more peaked, pushing it away from a uniform distribution and thus avoiding high-entropy collapse.

**Cropping Augmentation.** In principle, one can omit view-invariance by using identical inputs—reducing the objective to some form of mutual information maximization with centering and sharpening; this does not necessarily lead to useful representations [55, 56]. Instead, strong performance in self-supervised learning is often tied to carefully designed view augmentations. However, rather than relying on hand-crafted occlusions or corruptions [20, 57], we follow recent image-based SSRL trends that favor modality-agnostic augmentations such as masking or cropping [30, 32]. Specifically, AsymDSD generates crops by sampling randomly rotated bounding boxes with variable aspect ratios, rescaled to retain a random fraction $c$ of the original points (see Fig. 3).

**Multi-Crop.** Furthermore, we adopt *multi-crop* [35] to point clouds by generating two global crops $x_1^g$ and $x_2^g$ that cover a large fraction of the point cloud ($c > 0.4$), along with several local crops $x_i^l$ that may include as little as 5% of the original points ($0.05 < c < 0.4$). To simplify the implementation, all crops are subsampled to a fixed number of points, with local crops containing one-quarter the points of global crops. This sets up a challenging objective that encourages learning globally consistent representations from highly localized views.

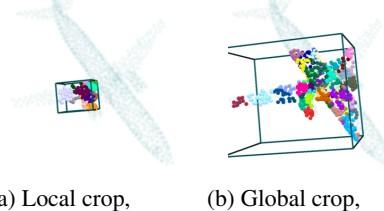

(a) Local crop, $c = 0.1$      (b) Global crop, $c = 0.6$

Figure 3: Local and global crops.

Particularly, we define the *multi-crop* loss over the full set of crops $\mathcal{V} = \left\{x_1^{\text{g}}, x_2^{\text{g}}, x_1^{\text{l}}, \ldots, x_{N_{\text{L}}}^{\text{l}}\right\}$ as:

$$\mathcal{L}^{\text{MC}}(\mathcal{V}) = \frac{1}{2|\mathcal{V}|} \sum_{u \in \left\{x_1^{\text{g}}, x_2^{\text{g}}\right\}} \sum_{v \in \mathcal{V}\setminus\{u\}} \mathcal{L}^{\text{CLS}}(u, v), \tag{2}$$

$$\mathcal{L}^{\text{CLS}}(u, v) = -P_{\theta'}^t (u)^{\intercal} \log P_\theta^s(v), \tag{3}$$

where $P_{\theta'}^t (u)$, and $P_\theta^s(v)$ are the probability mass vectors. These are obtained by attaching a dedicated CLS-token to each view to aggregate global shape information. The logits over the latent are computed via a separate projection module [11] to process this global representation: $f_\theta(u) = h_\omega^{\text{proj}}(f_\phi^{\text{enc}}(u)_{\text{CLS}})$. Additionally, a KoLeo loss [58, 18] is added to further encourage diversity in representations within a batch.

### 3.2 Patch-Level Objective: Masked Point Modeling (MPM)

While the global objective encourages inter-instance invariance and discrimination, the local objective promotes intra-instance predictability. Specifically, we adopt a masked point modeling (MPM) objective to learn spatially contextualized representations.

Given a *patchified* input consisting of $N_c$ patches, denoted by $\boldsymbol{x} = (\boldsymbol{X}^P, \boldsymbol{c})$, where $\boldsymbol{X}^P$ represents a collection of local point groups and $\boldsymbol{c}$ denotes the corresponding center points for these groups, we define a binary mask $m \in \{0, 1\}^{N_c}$ with a masking ratio $M_r$. The set of masked patch indices is given by $\mathcal{M} = \{i \mid m_i = 1\}$, and the set of visible patch indices by $\tilde{\mathcal{M}} = \{i \mid m_i = 0\}$. With latent MPM, the model trained on maximizing the log-posterior where the conditional comprises the visible context $\tilde{\boldsymbol{x}} = \left(\boldsymbol{X}_{\tilde{\mathcal{M}}}^P, \boldsymbol{c}_{\tilde{\mathcal{M}}}\right)$ accompanied by a position query $\mathbf{c}_i$ such that $i \in \mathcal{M}$, i.e.

$$\mathbb{E}_{(\mathbf{x}, z_i) \sim q_\phi(\mathbf{x}, z_i), \mathcal{M}} \left[ \sum_{i \in \mathcal{M}} \log p_\theta(z_i \mid \tilde{\mathbf{x}}, \mathbf{c}_i) \right]. \tag{4}$$

In simple terms, this expectation measures how well the model can predict the latent variable $z_i$ corresponding to the center $\boldsymbol{c}_i$ given a visible context $\tilde{\boldsymbol{x}}$. However, we cannot jointly sample $\boldsymbol{x}$ and $z_i$ as we do not have a model $q_\phi(\mathbf{x}, z_i)$. While variational methods such as BEiT [46] and Point-BERT [21] are enticing due to their mathematical interpretation [59], it remains guided by reconstruction-based learning, compromising semantic abstraction. Instead, we adopt a dynamic momentum teacher $p_{\theta'}^t$ similar to the global objective, allowing us to reformulate MPM as:

$$R^{\text{MPM}}(\theta, \theta') = \mathbb{E}_{\mathbf{x} \sim p(\mathbf{x}), \mathcal{M}} \left[ -\sum_{i \in \mathcal{M}} \mathbb{E}_{z_i \sim p_{\theta'}^t(z_i|\mathbf{x})} \left[ \log p_\theta^s(z_i \mid \tilde{\mathbf{x}}, \mathbf{c}_i) \right] \right] \tag{5}$$

In practice, the targets $z_i \sim p_{\theta'}^t(z_i \mid \mathbf{x})$ are not sampled from the teacher posterior as we use a latent with finite support. This enables the enumeration of all latent realizations for the exact computation of the cross-entropy as well as centering and sharpening to avoid collapse. Still, the design of the parameterized student $f_\theta^s$ and teacher $f_{\theta'}^t$ model underlying their respective posterior density function is crucial in ensuring a latent that represents semantically abstract information under the objective $R^{\text{MPM}}$. While symmetric architectures (e.g., denoising encoders [33, 30]) are plausible, we adopt an asymmetric design with the student hosting an encoder-predictor setup:

$$f_\theta^s(\tilde{\boldsymbol{x}}, \boldsymbol{c}_\mathcal{M}) \triangleq \left(h_\omega^{\text{proj}} \circ g_\psi^{\text{pred}}\right)\left(f_\phi^{\text{enc}}(\tilde{\boldsymbol{x}}), \boldsymbol{c}_\mathcal{M}\right), \tag{6}$$

$$f_{\theta'}^t(\boldsymbol{x}) \triangleq \left(h_{\omega'}^{\text{proj}} \circ f_{\phi'}^{\text{enc}}\right)(\boldsymbol{x}), \tag{7}$$

where $f^{\text{enc}}$ is a contextualizing encoder processing the visible context $\tilde{\boldsymbol{x}}$; $g^{\text{pred}}$ a prediction module that predicts the contextualized embeddings for each mask query $\bar{e}_i = \left(e^{\text{MASK}}, e_i^{\text{pos}}\right)$ given the encoded visible context. $e^{\text{MASK}}$ is a learnable mask token, and $e_i^{\text{pos}}$ the encoded position signal $\boldsymbol{c}_i$; $h^{\text{proj}}$ is a projection head, that separately projects each patch embedding to the discrete latent $z_i$.

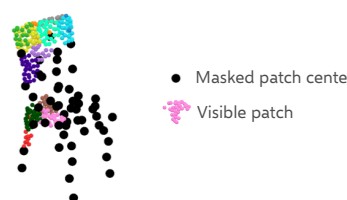

Figure 4: Leakage of the coarse shape via the positions of masked patches.

We consider the following benefits of this asymmetric design: (1) without the predictor, positional queries $c_\mathcal{M}$ inadvertently leak global spatial structure at the encoder stage (as illustrated in Fig. 4).

By deferring these queries to the predictor they can be effectively 'masked' by disabling attention between them. (2) Our encoder does not process masked embeddings during SSRL. This reduces the distribution mismatch between pre-training and downstream tasks, improving transferability. (3) The asymmetry due to a student predictor mitigates representations collapse and enhances training stability [60, 32]. Without this, the model may learn trivial solutions ignoring any contextual information. For example, a partitioning of space such that the centers $c_i$ are uniformly projected across spatial bins $\mathcal{Z}$ would maximize the marginal entropy $H\left(p_\theta(z)\right)$, rendering centering ineffective in overcoming this form of collapse. (4) Due to high mask ratios, the compute heavy encoder only processes a small number of visible patches. By contrast, the predictor that processes all the masked patches can be lightweight, thereby greatly increasing training efficiency.

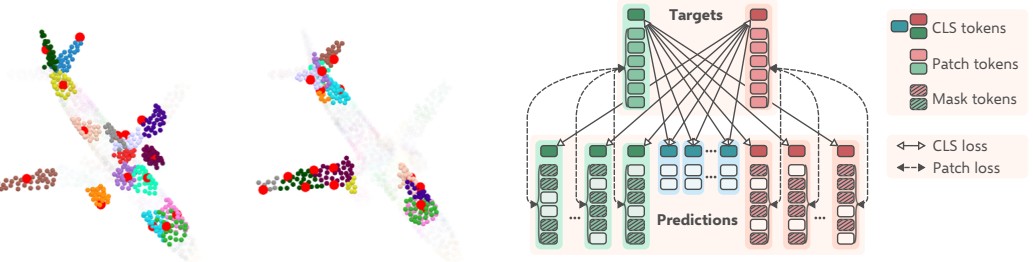

(a) Uniform      (b) Inverse block-wise

Figure 5: Masking strategies.

Figure 6: The losses of AsymDSD with multi-mask and multi-crop with both global (red and green) and local (blue) crops.

**Masking Strategy.** When it comes to the masking strategy, we observe that uniform masking (Fig. 5a) makes it generally easy to infer global structure due to the wide spatial distribution of visible patches. To address this, we implement inverse block-wise masking (Fig. 5b), which retains only a few small contiguous regions. This forces the model to infer the global shape from finer-grained details in a localized area. Specifically, we sample multiple fixed-sized blocks to add to the visible context via $k$-NN on center points $c$. To account for possible block overlap, the final mask is adjusted by randomly flipping masked or unmasked bits to achieve the target mask ratio.

**Multi-Mask.** Although the teacher $f_{\theta'}^t$ only performs a forward pass, it is comparatively costly due to full-context encoding. To amortize this cost, a *multi-mask* strategy is applied, where multiple masks $m^{(i)}$ are sampled per input $x$, averaging the MPM loss across them. This increases the effective batch size at a fraction of the usual cost, as the teacher targets as well as the non-contextualized student patch embeddings can be reused across masks.

### 3.3 Dual Objective Learning

AsymDSD unifies global invariance learning and local masked point modeling (MPM) within a single framework. Since both objectives operate in a latent space, it enables parameter sharing without architectural entanglement. Specifically, the encoder $f_\phi^{enc}$ is shared across objectives, while separate projection heads are maintained for the global and local tasks to accommodate objective-specific dynamics with minimal computational overhead. The predictor $g_\psi^{pred}$ is not used to refine the CLS token, meaning it is exclusive to the MPM branch. *multi-mask* is applied to the two global crops, whereas the local crops remain unmasked. These masked global views create additional cross-view comparisons for the global objective, and can be seen as a form of implicit denoising, similar to MSN [19]. Figure 6 summarizes the interactions between the outputs in the complete framework.

## 4 Experiments

We evaluate AsymDSD through extensive experiments across 3D recognition, few-shot classification, and part segmentation, including studies on scalability and ablations.

### 4.1 ShapeNet Pre-Training

**Implementation Details.** We follow the common SSRL pre-training protocol involving ShapeNet-Core [36], consisting of 41,952 CAD models across 55 categories. Point clouds are generated by

Table 1: Overall accuracy on **ModelNet40** and **ScanObjectNN**. Where available, the accuracy without voting is reported. ST indicates a ViT-S sized standard transformer as described in Section A and Table 7c; SM indicates single-modal training; **#P(M)** indicates the number of parameters in millions.

| Method | Reference | #P(M) | ST | SM | ModelNet40 | ScanObjectNN | | |
| --- | --- | --- | --- | --- | --- | --- | --- | --- |
| | | | | | | OBJ_BG | OBJ_ONLY | PB_T50_RS |
| *Supervised Learning Only* | | | | | | | | |
| PointNet | CVPR '17 [62] | 3.5 | × | ✓ | 89.2 | 73.3 | 79.2 | 68.0 |
| PointMLP | ICLR '22 [63] | 12.6 | × | ✓ | **94.1** | - | - | 85.4±.3 |
| PointNeXt | NeurIPS '22 [64] | 1.4 | × | ✓ | 94.0 | - | - | **87.7±.4** |
| **Adapted ViT-S** | Appendix A | 22.1 | ✓ | ✓ | 92.9 | 87.6 | 86.7 | 83.5 |
| *Self-Supervised Representation Learning - Full Fine-tune* | | | | | | | | |
| Point-BERT | CVPR '22 [21] | 22.1 | ✓ | ✓ | 93.2 | 87.43 | 88.12 | 83.07 |
| Point-MAE | ECCV '22 [22] | 22.1 | ✓ | ✓ | 93.2 | 90.02 | 88.29 | 85.18 |
| PointGPT-S | NeurIPS '23 [24] | 22.1 | ✓ | ✓ | 94.0 | 91.6 | 90.0 | 86.9 |
| **AsymSD-CLS-S** | Section 3.1 | 22.1 | ✓ | ✓ | 93.6 | **92.77** | **90.53** | **88.72** |
| **AsymSD-MPM-S** | Section 3.2 | 22.1 | ✓ | ✓ | 94.0 | **92.77** | **91.39** | **88.58** |
| **AsymDSD-S** | Section 3.3 | 22.1 | ✓ | ✓ | **94.1** | **94.32** | **91.91** | **90.53** |
| MaskPoint | ECCV '22 [65] | - | × | ✓ | 93.8 | 89.3 | 88.1 | 84.3 |
| Point-M2AE | NeurIPS '22 [23] | 15.3 | × | ✓ | 94.0 | 91.22 | 88.81 | 86.43 |
| ReCon SM | ICML '23 [26] | 43.6 | × | ✓ | 93.6 | 94.15 | 93.12 | 89.73 |
| Point-RAE | ACMMM '23 [66] | 29.2 | × | ✓ | 94.0 | **95.53** | **93.63** | 90.28 |
| Point-FEMAE | AAAI '24 [25] | 27.4 | × | ✓ | 94.0 | 95.18 | 93.29 | 90.22 |
| PointMamba | Neurips '24 [67] | 12.3 | × | ✓ | 93.6 | 94.32 | 92.60 | 89.31 |
| *Self-Supervised Representation Learning - Linear* | | | | | | | | |
| Point-BERT | CVPR '22 [21] | 22.1 | ✓ | ✓ | 91.09±.15 | 84.17±.30 | 87.19±.16 | 74.44±.12 |
| Recon MAE | ICML '23 [22] | 43.3 | × | ✓ | 90.22±.09 | 82.77±.30 | 83.23±.16 | 74.13±.21 |
| point2vec | GCPR '23 [34] | 22.1 | ✓ | ✓ | 92.44±.04 | 82.75±.54 | 85.44±.26 | 74.25±.11 |
| Point-RAE | ACMMM '23 [66] | 28.9 | × | ✓ | - | 86.15±.33 | 86.31±.23 | 78.25±.30 |
| **AsymSD-CLS-S** | Section 3.1 | 21.8 | ✓ | ✓ | 91.78±.10 | **90.67±.31** | **88.31±.31** | **83.19±.14** |
| **AsymSD-MPM-S** | Section 3.2 | 21.8 | ✓ | ✓ | **93.55±.05** | 89.00±.20 | 87.80±.14 | 81.04±.14 |
| **AsymDSD-S** | Section 3.3 | 21.8 | ✓ | ✓ | 92.52±.15 | 89.95±.21 | 88.73±.23 | 83.33±.14 |
| ACT | ICLR '23 [2] | 21.8 | ✓ | × | 91.36±.17 | 85.20±.83 | 85.84±.15 | 76.31±.26 |
| ReCon | ICML '23 [22] | 43.3 | × | × | 92.47±.22 | 89.50±.20 | **89.72±.17** | 81.36±.14 |

uniformly sampling 16,384 surface points per mesh and normalizing them to the unit sphere. For each input, we sample two global crops (1,024 points, 64 patches) and four local crops (256 points, 16 patches), with each patch comprising the $K = 32$ nearest neighboring points. Global crops are masked using inverse block-wise masking at a 70% ratio with four masks per crop. The encoder is a ViT-S backbone with RMSNorm and GELU, and the student predictor is a lighter ViT-Ti [61] variant with 6 layers. The projection heads expand embeddings to a 4096-way latent space. Pre-training is run for 300 epochs using AdamW, with a cosine learning rate schedule peaking at $5.0 \times 10^{-4}$, and a cosine EMA decay increasing from 0.995 to 1.0 during training. With a single RTX 4090, this takes roughly 18 hours to complete. For additional details, we refer the reader to Appendix B.1.

**Object Recognition.** We evaluate the downstream performance of AsymDSD on standard 3D object classification benchmarks using three protocols: *From Scratch*, *Linear*, and *Full Fine-tune* [2]. The teacher encoder (i.e., without projection head) is used for all downstream tasks, as it generally outperforms the student. In the *Linear* protocol, we freeze the encoder's weights and add a trainable linear layer atop the encoder. For the *Full Fine-tune* and *From Scratch* settings, we employ a 3-layer MLP head and update all model parameters during training. The inputs to these classification heads are formed by concatenating the CLS-token embedding with the mean and max pooled patch embeddings. All models are trained for 150 epochs using cross-entropy loss with label smoothing set to 0.2, a batch size 32, and a drop path rate of 0.2 [68]. The MLP head uses hidden dimensions of 256, batch normalization, and dropout ($p = 0.5$). Results to these experiments are shown in Table 1.

On **ModelNet40** [38], a clean synthetic dataset of 12k CAD models across 40 categories, **AsymDSD-S** achieves 94.1% accuracy with *Full Fine-tune*, a +1.2% increase compared to training *From Scratch* (**Adapted ViT-S**). Even with a simple linear probe, performance remains strong, indicating strong off-the-shelf representations. In contrast, point-reconstruction methods like MAE underperform by over 2% in the *Linear* setup. On **ScanObjectNN** [37], a real-world scanned object dataset, AsymDSD achieves 90.53% (+7.0%) on the hardest PB_T50_RS split, surpassing all prior methods with a standard transformer by +3.6%. While some methods slightly outperform AsymDSD on

cleaner subsets, they rely on architectural modifications with additional trainable parameters during fine-tuning. In fact, with equal trainable parameters under *Linear* probing, our method outperforms all other self-supervised approaches, including cross-modal methods like ACT [2] and ReCon [26].

**Few-Shot Classification.** We evaluate few-shot performance on ModelNet40 following the $m$-way, $n$-shot protocol of [69]. The model is fine-tuned with a high learning rate ($1 \times 10^{-3}$) and predictions are based on concatenated mean and max pooled patch embeddings (ignoring the CLS token). As shown in Table 2, **AsymDSD-S** achieves the best result in three out of four configurations, further supporting the off-the-shelve quality of the learned representations in low-data regimes.

**Part Segmentation.** We also evaluate semantic segmentation on ShapeNet-Part [70]. For this, we employ a PointNet++-style decoder [71] aggregating features from multiple encoder layers, following Point-MAE [22]. As shown in Table 3, **AsymDSD-S** achieves competitive mIoU scores, similar to transformer-based methods like Point-MAE [22] and PointGPT-S [24]. While methods such as point-M2AE outperform in this task, they rely on a U-Net-style architecture, which is generally better suited for dense prediction.

Table 2: Average overall accuracy and standard deviation on **ModelNet40 few-shot** over 10 independent runs per experiment.

| Method | ST | 5-way | | 10-way | |
|---|---|---|---|---|---|
| | | 10-shot | 20-shot | 10-shot | 20-shot |
| Point-BERT [21] | ✓ | 94.6±3.1 | 96.3±2.7 | 91.0±5.4 | 92.7±5.1 |
| MaskPoint [65] | × | 95.0±3.7 | 97.2±1.7 | 91.4±4.0 | 93.4±3.5 |
| Point-MAE [22] | ✓ | 96.3±2.5 | 97.8±1.8 | 92.6±4.1 | 95.0±3.0 |
| Point-RAE [34] | × | **97.3±1.6** | 98.7±1.3 | 93.3±4.0 | **95.8±3.0** |
| Point-FEMAE [34] | × | 97.2±1.9 | 98.6±1.3 | 94.0±3.3 | **95.8±2.8** |
| **AsymDSD-S** | ✓ | 96.1±2.8 | **98.8±1.3** | **94.4±3.5** | 95.8±3.3 |

Table 3: Part segmentation results on **ShapeNet-Part**. The mean IoU is over all instances (Inst.) or per-class (Cls.).

| Method | ST | mIoU | |
|---|---|---|---|
| | | Inst. | Cls. |
| Point-BERT [21] | ✓ | 85.6 | 84.1 |
| Point-MAE [22] | ✓ | **86.1** | **84.4** |
| PointGPT-S [24] | ✓ | 86.2 | 84.1 |
| **AsymDSD-S** | ✓ | 86.0 | **84.4** |
| point-FEMAE [25] | × | 86.3 | **84.9** |
| point-M2AE [23] | × | **86.5** | **84.9** |

## 4.2 Scaling Beyond ShapeNet

While ShapeNet is a standard benchmark for 3D SSRL, it is small by today's standards and only comprises synthetic data, constraining the quality of learned representations. To explore the scalability of AsymDSD, we pre-train on a substantially larger and more diverse dataset composed of synthetic and real-world 3D sources. This *Mixture* dataset combines a 10-dataset aggregate (133K instances) with Objaverse [4] (797K instances). In addition, we also explore a larger ViT-B backbone (**AsymDSD-B***), extending beyond the original ViT-S configuration (**AsymDSD-S***). Details of the dataset composition and training setup are provided in Appendix C.1.

Experiments show strong improvements from scaling (Table 4). When moving from ShapeNet to the more extensive *Mixture* dataset, fine-tuning accuracy improves by +0.6% on ModelNet40 and +3.19% on ScanObjectNN, surpassing the previous best PointGPT-L. Linear evaluation results show even greater gains, with improvements of +1.24% and +8.63%, respectively. We further compare with models that have undergone extensive post-pretraining on supervised data (PP) [24], including

Table 4: Results from scaling up pre-training. PP indicates additional post-pretraining with supervised data; CM cross-modal training; **#P(M)** the parameters count in millions.

| Method | Reference | #P(M) | PP | CM | ModelNet40 | ScanObjectNN | | |
|---|---|---|---|---|---|---|---|---|
| | | | | | | OBJ_BG | OBJ_ONLY | PB_T50_RS |
| | | | | | *Full Fine-tune* | | | |
| PointGPT-B | NeurIPS '23 [24] | 92.0 | × | × | 94.2 | 93.6 | 92.5 | 89.6 |
| PointGPT-L | NeurIPS '23 [24] | 310.0 | × | × | 94.5 | 95.7 | 94.1 | 91.1 |
| **AsymDSD-S*** | - | 22.1 | × | × | 94.4 | **97.07** | **94.83** | 92.89 |
| **AsymDSD-B*** | - | 92.1 | × | × | **94.7** | 96.73 | 94.32 | **93.72** |
| Point-MAE-B★ | NeurIPS '23 [24] | 120.1 | ✓ | × | 94.2 | 94.2 | 93.9 | 90.2 |
| PointGPT-B★ | NeurIPS '23 [24] | 92.0 | ✓ | × | 94.4 | 95.8 | 95.2 | 91.9 |
| ReCon++-B★ | ECCV '24 [72] | 177.4 | ✓ | ✓ | 94.6 | **98.62** | **96.21** | 93.34 |
| | | | | | *Linear* | | | |
| **AsymDSD-S*** | - | 21.8 | × | × | **93.98±.14** | 95.22±.17 | **94.51±.18** | 91.31±.09 |
| **AsymDSD-B*** | - | 91.8 | × | × | 93.76±.13 | **96.16±.11** | 93.44±.16 | **91.96±.14** |

Table 5: Ablations. Accuracy on ScanObjectNN is reported using a linear SVM or *Full Fine-tune*.

(a) Cropping and *multi-crop*.

| Method | SVM | FFt |
|---|---|---|
| **CLS-S** | **82.27** | **88.72** |
| — cropping | 73.60 ↓ **8.67** | 82.13 ↓ **6.59** |
| — *multi-crop* | 78.38 ↓ **3.89** | 85.64 ↓ **3.08** |

(b) Mask strategies.

| Method | Ratio | SVM | FFt |
|---|---|---|---|
| Inverse bw. | 0.6 | 79.63 ↓ 1.43 | 88.02 ↓ 0.46 |
| | 0.7 | 80.95 ↓ 0.11 | 88.20 ↓ 0.38 |
| **MPM-S** | 0.8 | 81.06 | **88.58** |
| | 0.85 | **81.37** ↑ 0.29 | 88.17 ↓ 0.41 |
| Uniform | 0.8 | 79.74 ↓ 1.32 | 87.99 ↓ 0.59 |

(c) Predictor. c: visible context; m: mask tokens

| Method | Attention | | Mem. | It/s | SVM | FFt |
|---|---|---|---|---|---|---|
| | Self | Cross | | | | |
| **MPM-S** | × | c | 16.2 | 4.80 | **81.06** | **88.58** |
| | c+m | × | 17.9 | 4.32 | 77.00 ↓ **4.06** | 85.98 ↓ **2.60** |
| — predictor | × | × | 23.3 | 3.11 | 24.95 ↓ **54.51** | 69.74 ↓ **18.84** |

(d) *multi-mask*. **Bs.**: Batch size; **Mm.**: Number of masks

| Method | Bs. | Mm. | Mem. | It/s | SVM | FFt |
|---|---|---|---|---|---|---|
| **MPM-S** | 128 | 8 | 16.2 | 4.80 | **81.06** | **88.58** |
| — *multi-mask* | 1024 | 1 | 48.8 | 1.46 | 81.16 ↑ 0.10 | 88.48 ↓ 0.10 |

the significantly larger cross-modal model ReCon++-B [26]. Remarkably, **AsymDSD-B***, despite its smaller size and purely self-supervised training, outperforms these models on both ModelNet40 and the hardest ScanObjectNN benchmark. These results underscore the effectiveness of AsymDSD in leveraging large-scale unlabeled 3D data to learn highly transferable representations.

### 4.3 Properties and Ablations

**Synergistic Objectives.** AsymDSD jointly optimizes two complementary objectives to enhance representation learning. To assess the synergistic effect, we train models on each objective separately with re-tuned hyperparameters. As shown in Table 1, both invariance learning (**AsymSD-CLS-S**) and masked point modeling (**AsymSD-MPM-S**) demonstrate competitive down-stream performance, but their combination (**AsymDSD-S**) exceeds both objectives across all benchmarks after fine-tuning. For linear probing, however, individual objectives may outperform due to task-specific inductive biases favoring certain benchmarks. To further understand this synergy, we examine the attention distance patterns (Figure 7) and observe that our latent MPM demonstrates an attention specialization pattern relatively aligned with the CLS objective. This contrasts with the typical inverted pattern found in masked autoencoders [26]. We hypothesize that this alignment enhances their composability.

**Ablations.** We highlight several ablations to demonstrate that the main contributions are not some simple add-ons for minor incremental gains, but are core to their objectives. As shown in Table 5a, cropping plays a key role in learning effective representations, with local crops (*multi-crop*) enabling full performance. We also evaluate different masking strategies (Table 5b), finding that our proposed inverse block-wise masking outperforms uniform masking. The method is robust to exact configurations, though best results are observed with masking ratios of 0.8 for MPM and 0.7 for the dual objective. Furthermore, Table 5c shows that the predictor is necessary to overcome collapse. That said, we observe that combining MPM with the global objective is sufficient to stabilizes training. Still, performance is reduced if global shape leakage is not addressed at the predictor (Appendix E). This is further evidenced by the performance gap between our efficient cross-attention-only predictor and the more expressive design with self-attention over all patches including masks (c+m). When we disable *multi-mask*, the efficiency of the design becomes especially obvious (Table 5d). There is virtually no performance degradation when using *multi-mask* with similar effective batch size, while reducing the memory usage and throughput by nearly 70%.

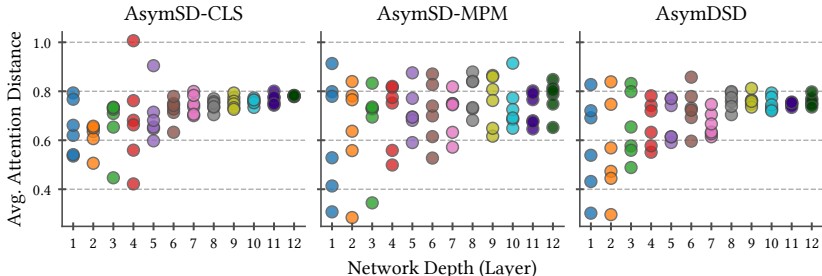

Figure 7: The average attention distance per attention head across the depth of the encoder.

Table 6: Comparison of prediction targets.

| Method | Raw points | Latent | ModelNet40 | | ScanObjectNN | |
|---|---|---|---|---|---|---|
| | | | SVM | FFt | SVM | FFt |
| CLS | - | - | 91.78 | 93.64 | 82.27 | 88.72 |
| MAE | ✓ | × | 92.91 | **94.04** | 78.49 | 87.75 |
| MPM | × | ✓ | **93.31** | **94.04** | **81.06** | **88.58** |
| CLS + MAE | ✓ | × | 91.90 ↑ 0.12 | 93.48 ↓ 0.16 | **82.93** ↑ 0.66 | 88.55 ↓ 0.17 |
| **CLS + MPM** | × | ✓ | **93.03** ↑ 1.25 | **94.13** ↑ 0.49 | 82.34 ↑ 0.07 | **90.53** ↑ 1.81 |

**Latent targets.** Table 6 highlights the impact of shifting the masked modeling targets from the input space (raw points) to the latent space. Our implementation of MAE shows substantial gains over Point-MAE [22], which can be attributed to mitigating global shape leakage through our predictor design and to the scale invariance introduced by variable-sized global crops. This reveals that some of our contributions extend beyond our framework. However, more importantly, as hypothesized, combining global invariance learning (CLS) with MAE (CLS + MAE) provides limited additional benefit and even results in a slight performance drop under full fine-tuning. In contrast, integrating CLS with our latent MPM objective (CLS + MPM) produces a clear synergistic effect, where the model to surpass both objectives when applied independently.

## 5   Conclusion and Limitations

**Conclusion.** In this paper, we introduced AsymDSD, a unified self-supervised learning framework for 3D point clouds that integrates masked modeling and invariance learning through asymmetric dual self-distillation. By avoiding reconstruction-based targets and addressing critical issues like shape leakage and representation collapse, AsymDSD enables efficient and semantically rich representation learning. The asymmetric design not only stabilizes training but also improves computational efficiency by decoupling heavy encoding from lightweight prediction. Extensive experiments demonstrate SOTA performance on multiple benchmarks, with strong generalization in low-shot and large-scale settings.

**Limitations.** While AsymDSD demonstrates strong performance on a wide-range of experiments, there are some limitations to the current work: (1) Pre-training and evaluations have been primarily conducted on object-centered datasets; and (2) the encoder architecture uses a flat hierarchy. These limitations are closely related, as the current architecture does not scale well to larger point clouds typically found in scene-level data. However, AsymDSD can be extended to hierarchical encoders with multi-resolution representations, which are better suited for such settings. Building on the promising results of this study, we view this as a compelling direction for future work.

## Acknowledgments

This research was conducted as part of the first author's master's thesis at the University of Groningen, in collaboration with Clear Timber Analytics. The authors would like to thank Hamidreza Kasaei for academic guidance and Alex van Gelder at Clear Timber Analytics for their support and provision of resources that contributed to this work. We also thank the Center for Information Technology of the University of Groningen for providing access to the Hábrók high performance computing cluster.

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

# A  Overall Model Pipeline

Although many dedicated models architectures for point cloud data have been devised, there is a lack of a unified architecture akin to those established in NLP and 2D CV. In these areas, model architectures have converged over time to a few dominant designs [73, 74, 53]. However, in recent times, a line of work on 3D SSRL has emerged [22, 21, 34, 24] that relies on a relatively simple model design that integrates the standard transformer architecture [74, 53]. This model architecture is detailed in this section. A detailed overview of this pipeline with AsymDSD pre-training is shown in Figure 8.

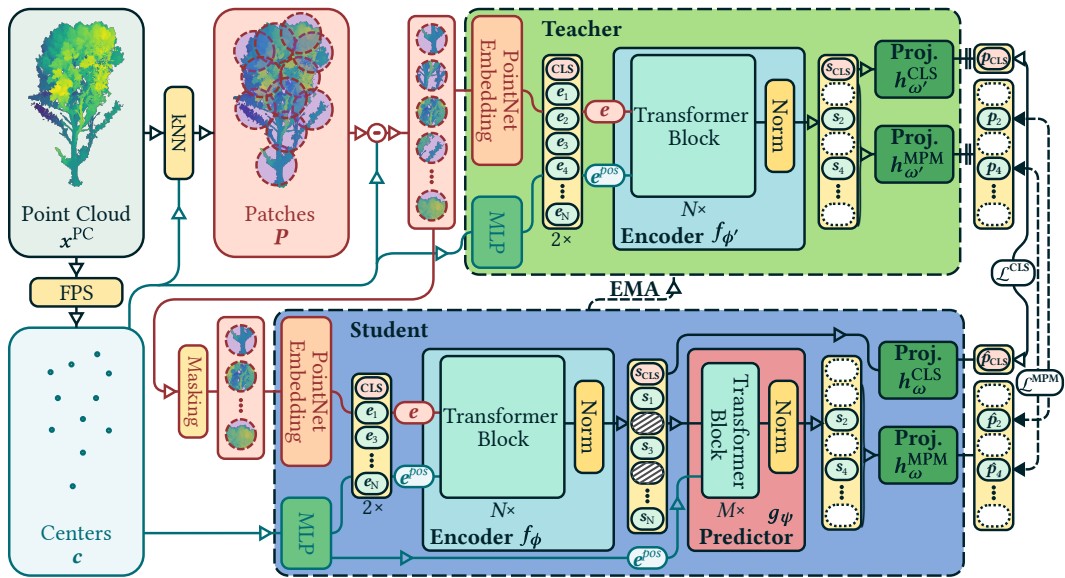

Figure 8: Overview of the processing pipeline for AsymDSD for a single point cloud through both the *Teacher* and *Student* network. The red colored arrows and modules indicate the stream of local structural information. The blue colors indicate the stream of global positional information. These streams get mixed at the encoder and predictor networks.

## A.1  Patch Tokenization

The first step of the processing pipeline involves the abstraction of the point set to a smaller set of point patches to obtain a manageable set of units for further processing. These patches are local groups of points that are comparable to image patches in vision transformers [53]. These patches are subsequently embedded to obtain a set of patch tokens.

### A.1.1  Input

The input is a point cloud $\mathcal{S}$, consisting of a finite collection of pairs of point positions $p^{(i)} \in \mathcal{P} \subset \mathbb{R}^3$ and $F$ point features $f^{(i)} \in \mathcal{F} \subset \mathbb{R}^F$:

$$\mathcal{S} = \left\{ \left( p^{(i)}, f^{(i)} \right) \mid p^{(i)} \in \mathcal{P}, f^{(i)} \in \mathcal{F}, i = 1, \ldots, n \right\}. \tag{8}$$

However, for simplicity we consider the point cloud $\mathcal{S}$ in tensor representation $\boldsymbol{x}^{\mathrm{PC}}$:

$$\boldsymbol{x}^{\mathrm{PC}} = [\boldsymbol{p} \mid \boldsymbol{f}] \in \mathbb{R}^{N \times (3+F)}, \tag{9}$$

with point positions $\boldsymbol{p} \in \mathbb{R}^{N \times 3}$ and accompanying features $\boldsymbol{f} \in \mathbb{R}^{N \times F}$.

### A.1.2  Patchify

To form the patches, first, $N_c$ center points $\boldsymbol{c}$ are sampled via farthest point sampling (FPS). FPS is a procedure that iteratively samples points that are most distant from the already sampled points,

starting with a randomly sampled point. Subsequently, $k$-nearest neighbour (KNN) is employed to find the $K$-nearest points in $\boldsymbol{x}^{\mathrm{PC}}$ based on their corresponding positions $\boldsymbol{p}$ from each center $\boldsymbol{c}$. This procedure is referred to as *patchify* and can be expressed mathematically as:

$$\boldsymbol{c} = \mathrm{FPS}(\boldsymbol{p}), \qquad\qquad \boldsymbol{c} \in \mathbb{R}^{N_c \times 3}; \tag{10}$$

$$\boldsymbol{P} = \mathrm{KNN}\left(\boldsymbol{c}, \boldsymbol{p}, \boldsymbol{x}^{\mathrm{PC}}; K\right), \qquad\qquad \boldsymbol{P} \in \mathbb{R}^{N_c \times K \times (3+F)}. \tag{11}$$

To disentangle the global positional information from the local structural information, the reference frame of each patch is translated to its respective center:

$$\boldsymbol{X}^{\boldsymbol{P}} = [\boldsymbol{P}_{xyz} - \boldsymbol{c} | \boldsymbol{P}_f], \qquad\qquad \boldsymbol{X}^{\boldsymbol{P}} \in \mathbb{R}^{N_c \times K \times (3+F)}; \tag{12}$$

with $\boldsymbol{P}_{xyz}$ the point positions, and $\boldsymbol{P}_f$ the point features of the patches.

This *patchify* procedure is reflected as a part of the complete processing pipeline for AsymDSD in Figure 8.

### A.1.3 Patch Embedding

The obtained patches $\boldsymbol{X}^{\boldsymbol{P}}$ are themselves small point clouds. Accordingly, before further processing, they must be projected to an embedding space. This embedding process effectively boils down to the compression of the point cloud into a single feature vector describing its shape. For this purpose a small *PointNet Embedding* model is used, as shown in Figure 9a.

This embedding model first projects the points in each patch to a higher $D^1_{\mathrm{patch}}$-dimensional space through a simple shared multi-layer perceptron (MLP), and takes the maximum of this projection over the $K$ points. This essentially partitions the Euclidean space into regions. To make the partitioning dependent on the contents of the point cloud, the maximum feature vector is concatenated to the projected point features before being projected and pooled once more to obtain the final patch embeddings $\boldsymbol{e}^{\boldsymbol{P}}$. In other words:

$$\boldsymbol{Z}^{\boldsymbol{P}} = \mathrm{MLP}_1(\boldsymbol{X}^{\boldsymbol{P}}), \qquad\qquad \boldsymbol{Z}^{\boldsymbol{P}} \in \mathbb{R}^{N_c \times K \times D^1_{\mathrm{patch}}}; \tag{13}$$

$$\boldsymbol{e}^{\boldsymbol{P}} = \max_j \left(\mathrm{MLP}_2\left(\left[\boldsymbol{Z}^{\boldsymbol{P}} \,\middle|\, \max_k\left(\boldsymbol{Z}^{\boldsymbol{P}}_{:,k}\right)\right]\right)_{:,j}\right), \qquad\qquad \boldsymbol{e}^{\boldsymbol{P}} \in \mathbb{R}^{N_c \times D_{\mathrm{embed}}}; \tag{14}$$

where $\mathrm{MLP}_i$ is a multi-layer perceptron consisting of two linear layers interjected with a normalization and non-linear activation function.

### A.1.4 Position Embedding

The separated positions of the patches are similarly prepared for downstream processing. In contrast to textual language and images where tokens or patches are associated with discrete positions, the center points of each point patch are embedded in a continuous $\mathbb{R}^3$ Euclidean space. While many different methods have been proposed to reinject the positional information of point clouds [75–77], a simple and efficient absolute positional embedding (APE) is used here. The embedding is defined by a learnable mapping $f^{\mathrm{pos}} : \mathbb{R}^3 \mapsto \mathbb{R}^{D_{\mathrm{embed}}}$, and is implemented with a simple two-layer MLP:

$$\boldsymbol{e}^{\mathrm{pos}} = \mathrm{MLP}(\boldsymbol{c}), \qquad\qquad \boldsymbol{e}^{\mathrm{pos}} \in \mathbb{R}^{N_c \times D_{\mathrm{embed}}}. \tag{15}$$

## A.2 Contextualizing Encoder

At the core of the pipeline is the transformer encoder, which contains the majority of the parameters and carries out the bulk of the processing work. This encoder utilizes global self-attention across the patch tokens, thereby incorporating shape information from the entire point cloud to obtain globally contextualized embeddings.

### A.2.1 Input

To obtain the input to the tranformer encoder, a learnable class embedding $\boldsymbol{e}^{\mathrm{CLS}} \in \mathbb{R}^{1 \times D_{\mathrm{embed}}}$ is prepended to the patch embedding tokens, yielding the input $\boldsymbol{z}^0$ of the encoder model:

$$\boldsymbol{z}^0 = \left[\boldsymbol{e}^{\mathrm{CLS}}; \boldsymbol{e}^{\boldsymbol{P}}\right], \qquad\qquad \boldsymbol{z}^0 \in \mathbb{R}^{(1+N_c) \times D_{\mathrm{embed}}}. \tag{16}$$

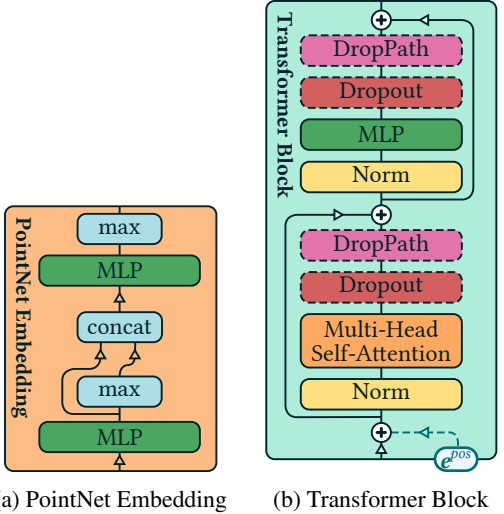

(a) PointNet Embedding    (b) Transformer Block

Figure 9: Building blocks of the processing pipeline.

This additional CLS token is not associated with a position in $\mathbb{R}^3$ and serves to build up a global representation, which is beneficial for non-localalized tasks such as classification or the global self-distillation objective of AsymDSD.

### A.2.2 Transformer Encoder

A standard transformer encoder [74] with pre-normalization is used, following the overall model structure of ViT [53]. It consists of a stack of $L$ transformer blocks (Figure 9b). Due to the expected importance of positional information of the patches, the position embedding is readded to the input of each block. This diverges from the typical approach where the position embedding is added once before the first block of the transformer. Notably, the model maintains a fixed embedding size throughout the depth of the network.

## B    Additional Implementation Details of AsymDSD

### B.1    ShapeNet Pre-Training

Pre-training details including pre-processing, model and training hyperparameters are provided in Table 7.

### B.2    Inverse Block-wise masking for point clouds

Block-wise masking has been explored for point cloud data in several studies. However, the common implementation involves sampling a single block [21, 22, 34]. AsymDSD generalizes this by sampling any number of blocks of a pre-determined size $B_s$ expressed in the number of patches. Furthermore, we consider inverse block-wise masking, following Baevski et al. [31], where the blocks instead indicate which patches to keep.

The difficulty with sampling multiple blocks is that they may partially overlap, which makes it difficult to mask the desired ratio of patches. To address this issue, the mask ratio is slight increased by adjust ratio $A_r$ [31] to increase the number of blocks $N_B$ to be sampled. Specifically:

$$N_B = \text{round}\left(N_c * \frac{(1 - M_r) + A_r}{B_s}\right), \tag{17}$$

where $N_B$ is the number of blocks, $N_c$ the number of patches, $M_r$ the mask ratio, $A_r$ the adjust ratio, and finally $B_s$ the block size.

The blocks are subsequently generated by sampling $N_B$ center positions, and accumulating the $B_s$ nearest patches according to the $L_2$-distance. This may lead to over- or under-masking depending

on the amount of overlap. However, this is resolved by randomly swapping the mask bits until the desired mask ratio is achieved. The adjust ratio $A_r$ can be chosen such that the number of swaps to be performed is minimized.

Table 7: AsymDSD's hyperparameters for ShapeNet pre-training.

(a) The data pre-processing parameters including cropping and masking.

| Parameter | Symbol | Value | |
|---|---|---|---|
| *Data Pre-processing* | | | |
| # Points | | 16 384 | |
| Augmentation-1 | | Rotate $z$-axis | |
| Augmentation-2 | | Anistropic Scaling [0.8, 1.2] | |
| Normalization | | Unit Sphere | |
| *Cropping* | | | |
| | | Global | Local |
| # Crops | $N_G, N_L$ | 2 | 4 |
| # Points | $N$ | 1024 | 256 |
| # Patches | $N_c$ | 64 | 16 |
| # Patch Points | $K$ | 32 | 32 |
| Crop Fraction | | [0.4, 1.0] | [0.05, 0.4] |
| *Masking* | | | |
| Mask Sampler | | Inverse block-wise | |
| Multi Mask | $N_{\mathrm{mm}}$ | 4 | |
| Mask Ratio | $M_r$ | 0.7 | |
| Block Size | $B_s$ | 6 | |
| Adjust Ratio | $A_r$ | 0.1 | |

(b) The training hyperparameters.

| Parameter | Symbol | Value |
|---|---|---|
| *Training* | | |
| Batch Size | $\mathcal{B}$ | 128 |
| Epochs | | 300 |
| Precision | | FP16 mixed [78] |
| *Optimizer* | | |
| Optimizer | | AdamW [79] |
| LR Schedule | $\lambda_{\mathrm{lr}}$ | Cosine Annealing |
| Base Learning Rate | | $5.0 \times 10^{-4}$ |
| # LR Warmup Epochs | | 10 |
| Momentum Decay | $\beta_1, \beta_2$ | 0.9, 0.999 |
| Weight Decay | | 0.05 |
| KoLeo Scale | $\alpha$ | 0.01 |
| Gradient Clip Norm | | 10.0 |
| *Self-Distillation* | | |
| EMA Schedule | $\eta$ | Cosine |
| EMA Start, End | $[\eta^0, \eta^E]$ | [0.995, 1.000] |
| Centering Momentum | $\eta_l$ | 0.9 |
| Student Temp | $\tau_s$ | 0.1 |
| Teacher Temp Schedule | | Linear Warmup |
| Teacher Temp CLS | $\tau_t^{\mathrm{CLS}}$ | [0.04, 0.07] |
| Teacher Temp Patch | $\tau_t^{\mathrm{patch}}$ | [0.05, 0.07] |
| # Teacher Warmup Epochs | | 10 |

(c) The hyperparameters and details of the model.

| Parameter | Symbol | Value |
|---|---|---|
| *Shared Model Defaults* | | |
| Normalization | | RMSNorm [80] |
| Activation | | GELU [81] |
| Linear Bias | | False |
| *Patch Embedding (Sec. A.1.3)* | | |
| $\mathrm{MLP}_1$ Dims | | 128, 256 |
| $\mathrm{MLP}_2$ Dims | | 512, 384 |
| *Position Embedding (Sec. A.1.4)* | | |
| MLP Dims | | 128, 384 |
| *Contextualizing Encoder (Sec. A.2)* | | |
| Transformer Block | | Transformer Encoder |
| Embedding Dim | $D_{\mathrm{embed}}$ | 384 |
| MLP Expansion Dim | | 1536 |
| # Layers | $L$ | 12 |
| # Attention Heads | $H$ | 6 |
| *Predictor* | | |
| Transformer Block | | Transformer Decoder |
| Embedding Dim | $D_{\mathrm{embed}}$ | 192 |
| MLP Expansion Dim | | 768 |
| # Layers | $L$ | 6 |
| # Attention Heads | $H$ | 3 |
| *Projection head* | | |
| MLP Dims | | 1024, 1024, 256 |
| Output Dim | $N_{\mathrm{tok}}$ | 4096 |
| Linear Bias | | True |

## C  Scaling Pre-Training

### C.1  Mixture Dataset

To scale the amount of training data, 3D models and scans from various sources were accumulated to make one large diverse datasets. Table 8 provides an overview of these datasets and their basic properties. The first 10 listed datasets comprise a total of 133 668 instances. When combined with the Objaverse dataset [4], they form *Mixture*, resulting in a total of 930 752 instances.

To provide some more details on the compilation process: for synthetic datasets without pre-existing point clouds, we uniformly sampled $16\,\mathrm{k}$ points from the surface of each mesh. In the case of scanned scene datasets with annotated objects—specifically S3DIS [82] and SUN RGB-D [83]—individual objects were cropped from the scenes and saved as individual instances. Notably, object instances with less than $2\,\mathrm{k}$ points were thrown away—these are predominantly of the category *clutter*. Additionally, all objects were rotated to their natural upright position in a shared reference frame. Any other features, such as color or normals, were not used.

## C.2 Training details

These models that are pre-trained on *Mixture* are referred to as **AsymDSD-S\*** for the ViT-S-sized model and **AsymDSD-B\*** for the ViT-B model. The architectural details of the larger model are shown in Table 10, with parameter counts for both models listed in Table 9. Notably, AsymDSD-B scales the patch embedding to accommodate the larger ViT-B contextualizing encoder, and upgrades the predictor from a ViT-Ti to a ViT-S, with half the usual number of layers.

AsymDSD-S\* was trained for 100 epoch on a batch size of 128, totaling $727\,\mathrm{k}$ optimization steps, in roughly 100 hours on a single A100 GPU. AsymDSD-B\* was trained for both 50 epochs, taking around 175 hours, respectively, on the same hardware.

## C.3 Evaluation on Objaverse-LVIS

The Objaverse dataset contains a subset of approximately $47\,\mathrm{k}$ objects annotated with one of the 1,156 categories from the LVIS dataset [84]. Unlike other datasets, it does not come with a predefined training or test split and is typically used for zero-shot evaluation in language-aligned models [85, 86]. Since our model does not produce language-aligned representations, we assess its representational quality through few-shot probing using both linear and kNN classifiers.

In particular, 10 instances are randomly sampled per category, and the remaining instances are added to the test set. We exclude any category with 10 or fewer instances, resulting in a total of 1060 remaining categories. Again, we exclude any additional features such as color and sample 1024 points per instance. The few-shot sampling strategy was repeated 10 times to remove most of the noise from sampling or training. The results from these experiments are shown in Table 11.

Table 8: Datasets for scaling pre-training. $^\dagger$ indicates that object instances are sampled from scenes.

| Dataset | # Instances | # Classes | Type |
|---|---|---|---|
| ShapeNetCore v2 [36] | 52 470 | 55 | Synthetic |
| 3D-FUTURE [87] | 16 560 | 50 | Synthetic |
| ScanObjectNN [37] | 16 034 | 15 | Scanned |
| ModelNet40 [38] | 9843 | 40 | Synthetic |
| S3DIS$^\dagger$ [82] | 8948 | 14 | Scanned |
| SUN RGB-D$^\dagger$ [83] | 8451 | - | Scanned |
| Amazon Berkeley Objects [88] | 7953 | - | Synthetic |
| OmniObject3D [89] | 5911 | 216 | Synthetic |
| Toys4K [90] | 4000 | 105 | Synthetic |
| Google Scanned Objects [91] | 1030 | - | Scanned |
| Objaverse [4] | 797 084 | 1156+ | Mixed |
| **Mixture** | 930 752 | - | **Mixed** |

Table 9: The total number of parameters of each module.

| Module | Symbol | # Parameters (M) | |
|---|---|---|---|
| | | AsymDSD-S | AsymDSD-B |
| Patch Embedding | $f_\phi^{\mathrm{embed}}$ | 0.5 | 6.5 |
| Transformer Encoder | $f_\phi^{\mathrm{ctx}}$ | 21.2 | 85.0 |
| Predictor | $f_\psi^{\mathrm{pred}}$ | 2.9 | 11.2 |
| Projection Head | $f_\omega^{\mathrm{proj}}$ | $2 \times 2.8$ | $2 \times 3.2$ |

Table 10: The hyperparameters and details of AsymDSD-B.

| Parameter | Value |
|---|---|
| *Patch Embedding (Sec. A.1.3)* | |
| $\mathrm{MLP}_1$ Dims | 128, 256, 512 |
| $\mathrm{MLP}_2$ Dims | 1024, 768 |
| *Position Embedding (Sec. A.1.4)* | |
| MLP Dims | 128, 768 |
| *Contextualizing Encoder (Sec. A.2)* | |
| Transformer Block | Transformer Encoder |
| Embedding Dim | 768 |
| MLP Expansion Dim | 3072 |
| # Layers | 12 |
| # Attention Heads | 6 |
| Drop Path [68] | 0.1 |
| *Predictor* | |
| Transformer Block | Transformer Decoder |
| Embedding Dim | 384 |
| MLP Expansion Dim | 1536 |
| # Layers | 6 |
| # Attention Heads | 6 |

Table 11: Top-$k$ few-shot performance on **Objaverse LVIS** subset. The number of shots per category is 10 and the average is reported over 10 independent runs. kNN uses the 5-nearest neighbors.

| Method | Linear | | | kNN | | |
|---|---|---|---|---|---|---|
| | Top1 | Top3 | Top5 | Top1 | Top3 | Top5 |
| **AsymDSD-S*** | 37.75 | 57.43 | 64.70 | 33.30 | 49.77 | 54.27 |
| **AsymDSD-B*** | **38.36** | **58.22** | **65.43** | **33.68** | **50.24** | **54.68** |

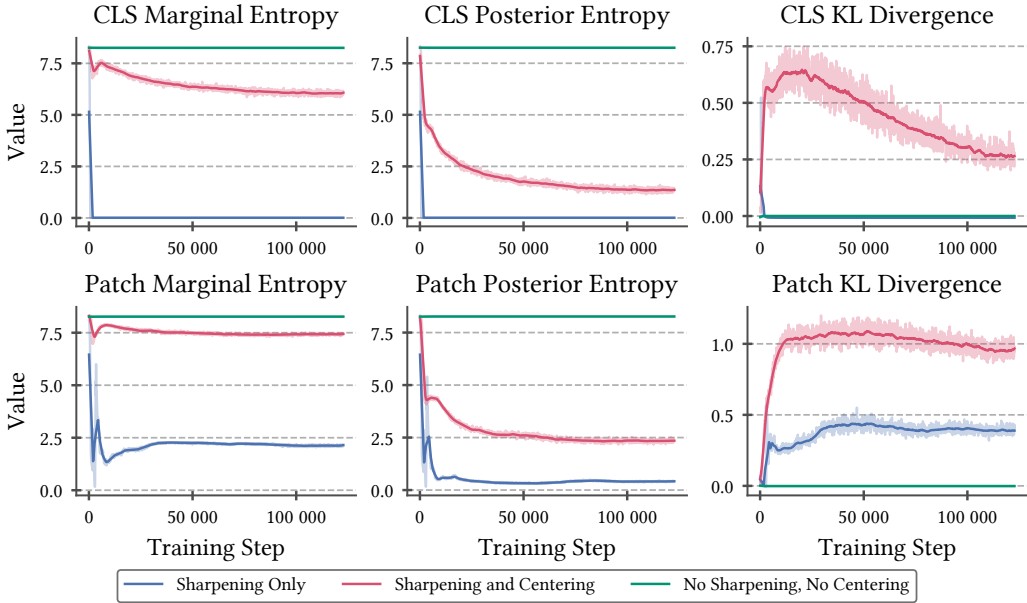

Figure 10: Marginal and posterior entropy, and KL divergence during training.

# D   Properties and Ablations of AsymDSD

This section presents additional experiments and results to gain deeper insights into the properties of AsymDSD.

## D.1   Student-Teacher dynamics

The dynamics that unfold between the student and teacher are central to the effectiveness of SSRL with AsymDSD. In particular, we demonstrate the occurrence of representation collapse and highlight the importance of centering and sharpening mechanisms in mitigating this issue. In addition, we look at the teacher performance relative to the student.

### D.1.1   Sharpening and Centering

For AsymDSD, sharpening and centering on the discrete targets is a central method of defense against the main modes of collapse. To demonstrate the effectiveness of these techniques, we trained AsymDSD without sharpening and centering, only with sharpening, and with both.

To determine wether collapse occurs in these setups, we compute the empirical marginal and posterior entropy over the batches $\mathcal{B}$ during training:

$$H\left(p_{\theta'}^t(\mathrm{z}_i)\right) \approx H\left(\sum_{x \in \mathcal{B}} p_{\theta'}^t(\mathrm{z}_i \mid x)\right), \qquad (18)$$

$$\mathbb{E}_{\mathrm{x} \sim p(\mathrm{x})}[H(p_{\theta'}^t(\mathrm{z}_i \mid \mathrm{x}))] \approx \sum_{x \in \mathcal{B}} H\left(p_{\theta'}^t(\mathrm{z}_i \mid \mathrm{x})\right), \qquad (19)$$

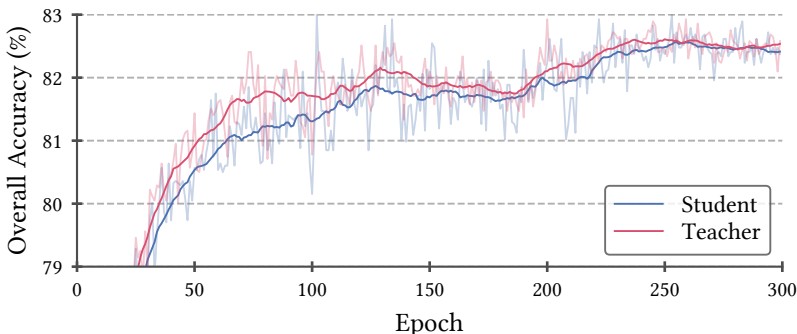

Figure 11: Student versus teacher performance. It shows plots of the overall accuracy on the hardest subset of ScanObjectNN with a linear SVM.

where $i = \text{CLS}$ or $i \in \mathcal{M}$ for the CLS-token or the patch tokens respectively. These metrics are plotted in Figure 10 alongside the KL divergence from the two training objectives $R^{\text{CLS}}(\theta, \theta')$ and $R^{\text{MPM}}(\theta, \theta')$. Note that we can simply decompose the cross-entropy to obtain the KL divergence: $D_{\text{KL}}(\text{x} \parallel \text{y}) = H(\text{x}, \text{y}) - H(\text{x})$.

Without sharpening and centering, the posterior collapses to the uniform distribution over both the CLS- and patch tokens, as indicated by the maximum entropy for $N_{\text{tok}} = 4096$ of $\log(4096) \approx 8,318$. On the other hand, when sharpening the targets, the marginal entropy falls to zero on the CLS-token, which indicates that the model always assigns a probability of 1.0 to the same token. This effect is not observed for the patch tokens, which demonstrate a non-zero marginal entropy larger than the posterior entropy. In this scenario we also observe non-zero KL divergence. We believe this greater stability on the patch tokens to be a result of the predictor module, which has been shown to be a key component in stabilizing training (Tab. 5c) [60, 32].

When combining sharpening with centering, as per AsymDSD, both modes of collapse are avoided, as the marginal entropy now stays high while the posterior entropy goes down during training, as desired. After all, the difference between the two quantifies the mutual information between the input x and latent $z_i$.

### D.1.2 Outperforming Teacher

A property that is sometimes witnessed with joint embedding architectures with a momentum teacher, is the teacher outperforming the student with the idea that the teacher guides the student towards higher quality representations [92, 11]. As shown in Figure 11, AsymDSD demonstrates this effect with the teacher on average outperforming the student during training when evaluated with a linear SVM on ScanObjectNN. That said, this performance difference was not observed on the ModelNet40 dataset. However, this might be related to fact that peak accuracy with a linear SVM on ModelNet40 is obtained at around 10% of the total training duration.

### D.2 Representational Quality

For a qualitative assessment of the representations of the AsymDSD's pre-trained encoder, we project patch embeddings to RGB space with t-SNE. Specifically, patch embeddings of the AsymDSD-B* pre-trained encoder are computed for 200 samples from the same object category in the ModelNet40 dataset. Each point is colored according to the inverse distance weighted average of the three nearest RGB patch embeddings.

The visualizations are shown for three object categories in Figure 12. We also included visualizations from a model that is trained with supervised learning on the ModelNet40 dataset. It stands out that AsymDSD shows high congruency among the projected embeddings from patches that comprise a semantic part of an object. This uniformity is also manifested between parts from different objects, e.g. the cyan colored ears of the cups, or the green cone shaped lamp covers. Interestingly, the latter example of cone shaped lamp covers also demonstrates strong scale invariance. We believe that cropping and multi-crop plays a key role in learning this invariance, as it leads to varying patch

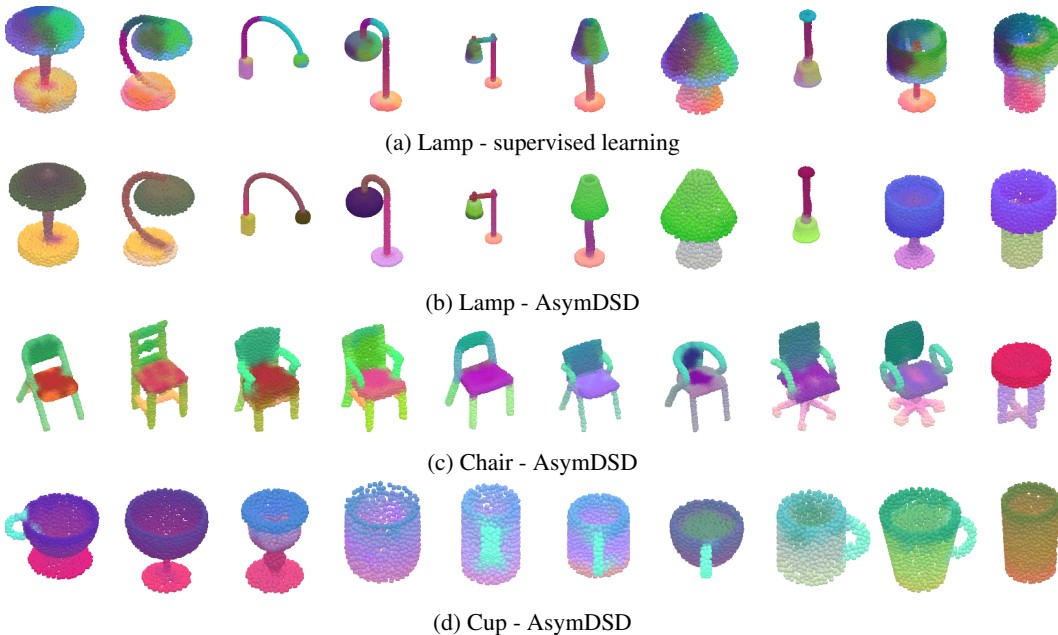

(a) Lamp - supervised learning

(b) Lamp - AsymDSD

(c) Chair - AsymDSD

(d) Cup - AsymDSD

Figure 12: Zero-shot coloring of points according to inverse distance weighted nearest patch embeddings from AsymDSD-B* projected to RGB space with t-SNE. The projection is computed with 200 instances from the same class with a perplexity of 30.

densities for the same object. Notably, these properties with strong semantic separation are not present with the model that is trained with standard supervision. This supervised model instead shows a strong positional bias among the projected embeddings.

# E  Additional Ablations

## E.1  Predictor in AsymDSD

Table 12 presents results from ablation experiments on the dual-objective formulation involving the predictor module. These results suggest that jointly optimizing both objectives improves robustness: even when the predictor is removed (cf. Table 5c), downstream performance does not collapse. Nonetheless, there is a modest drop in accuracy, and throughput decreases significantly. We hypothesize that this collapse-resistance stems from global invariance learning mitigating the model's collapse towards representing only the positional queries—given the variability of such queries across different crops.

Nevertheless, some performance degradation is expected when the predictor is removed due to leakage of the overall object shape through all mask queries. This effect should become pronounced when unmasked local crops are removed and the predictor is also excluded—i.e., when all instances expose a large portion of the coarse shape. Under this configuration, results indeed show a substantial drop in performance, most notably in linear SVM accuracy.

Table 12: Ablations on **AsymDSD-S**.

| Method | Fig. | MC | Attention | | Mem. | It/s | ModelNet40 | | ScanObjectNN | |
| | | | Self | Cross | | | SVM | FFt | SVM | FFt |
|---|---|---|---|---|---|---|---|---|---|---|
| **AsymSD** | 13c | ✓ | ✗ | c | 25.6 | 2.57 | **93.03** | **94.13** | **82.34** | **90.53** |
| | 13a | ✓ | c+m | ✗ | 26.8 | 2.42 | 92.67 ↓ 0.36 | 93.84 ↓ 0.29 | 82.13 ↓ 0.21 | 89.45 ↓ 1.08 |
| — predictor | - | ✓ | ✗ | ✗ | 31.9 | 2.07 | 92.87 ↓ 0.17 | 94.00 ↓ 0.13 | 81.82 ↓ 0.52 | 89.49 ↓ 1.04 |
| | 13c | ✗ | ✗ | c | 22.8 | 3.01 | 93.03 → 0.0 | 93.84 ↓ 0.29 | 81.99 ↓ 0.35 | 89.52 ↓ 1.01 |
| — predictor | - | ✗ | ✗ | ✗ | 29.3 | 2.51 | 92.18 ↓ **0.85** | 93.64 ↓ **0.49** | 79.88 ↓ **2.46** | 88.13 ↓ **2.40** |

In fact, when using a predictor and optimizing the joint objective, *multi-crop* may no longer be essential. In particular, inverse block-wise masking can simulate localized contexts similar to those provided by local crops, but without incurring the additional computational cost of processing them separately. However, under the current implementation, the variable nature of local crops allows for significantly smaller contexts compared to what is achievable through masking applied to global crops. This likely explains the observed performance improvements when local crops are included alongside the dual objective. However, these findings suggest a promising direction for future work: exploring the use of variable mask ratios as a potential alternative to multi-crop strategies.

## E.2 Objective Function

The MPM objective $R^{\mathrm{MPM}}$ is presented as a cross-entropy objective between the teacher and student posterior over the masked patches (Eq. 5). However, as demonstrated in other works [32], the high-dimensional student representations (i.e., before projection to discrete latent z) can also be directly regressed onto the teacher representations without representational collapse ensuing.

Table 13 presents results comparing these approaches. We find that direct regression with a smooth $L_1$-loss ($\beta = 2$) [93] without the KL divergence-based $R^{\mathrm{MPM}}$ objective—and thus no MI-maximizing measures on a discrete latent projection—does not lead to collapsed representations. However, we observe from hyper-parameters sweeps across several training runs that it is difficult to stabilize training. This is reflected in down-stream performance that eventually starts decreasing during training. Interestingly, when regression is combined with $R^{\mathrm{MPM}}$, it stabilizes training and improves performance across all benchmarks compared to training only on $R^{\mathrm{MPM}}$. We therefore use this setting combining the two losses in **AsymSD-MPM-S**. **AsymDSD-S** only uses cross-entropy minimization, as we did not observe down-stream improvement by adding an additional regression loss.

Table 13: Loss functions for MPM. `CE` indicates the cross-entropy objective $R^{\mathrm{MPM}}$. `RG` indicates a regression loss.

| CE | RG | ModelNet40 | | ScanObjectNN | |
|----|----|-----------|-----|-------------|-----|
| | | SVM | FFt | SVM | FFt |
| ✓ | ✗ | 92.10 ↓ 0.53 | 93.76 ↓ 0.28 | 78.97 ↓ 0.49 | 87.75 ↓ 0.83 |
| ✗ | ✓ | 90.51 ↓ **2.12** | 93.44 ↓ **0.60** | 69.74 ↓ **9.72** | 87.37 ↓ **1.21** |
| ✓ | ✓ | **92.63** | **94.04** | **79.46** | **88.58** |

### E.2.1 Additional Predictor Designs

The predictor module is a central component of MPM with Asymmetric Self-Distillation. It was argued that it not only enhances training efficiency but also improves the overall training architecture by mitigating issues such as global shape leakage, distribution mismatch, and representation collapse. These results were presented in Table 5c, but we experimented with several alternative predictor designs. The results of these experiments are presented in Table 14, with the designs shown in Figure 13.

While it was already shown that the typical transformer encoder implementation (Fig. 13a) leads to significantly reduce performance. We also tested cross-attention with concatenation of the query token itself (Fig. 13d), but observe that this yields no significant benefit, at the cost of higher memory usage and lower throughput.

**Multi-Block.** So far the two extremes of separately predicting all masked patches ($p_\theta(z_i \mid \bar{\mathbf{x}}, \mathbf{c}_i)$, $\forall i \in \mathcal{M}$) or predicting all masked patches at once ($p_\theta(z_i \mid \bar{\mathbf{x}}, \mathbf{c}_{\mathcal{M}})$, $\forall i \in \mathcal{M}$) have been explored. This can be generalized to predicting any number of masked patches at once [32]. Block-wise masking is particularly suitable alongside such generalized prediction objective, given that it provides local neighborhoods of masked regions to be 'unmasked'. In this way, the prediction is performed by providing the positional queries for a block of patches, allowing a coordinated build-up of representations over a region that extends beyond the bounds of a single patch without revealing the global shape of the object.

For *multi-block*, we use blocks with size $B_s = 9$, using two suitable designs for a coordinated build up, as shown in Figure 13b and 13e. Although these designs achieved strong downstream

performance, they offered no clear advantage over the simpler approach of predicting each patch token independently, while coming with the cost of lower throughput.

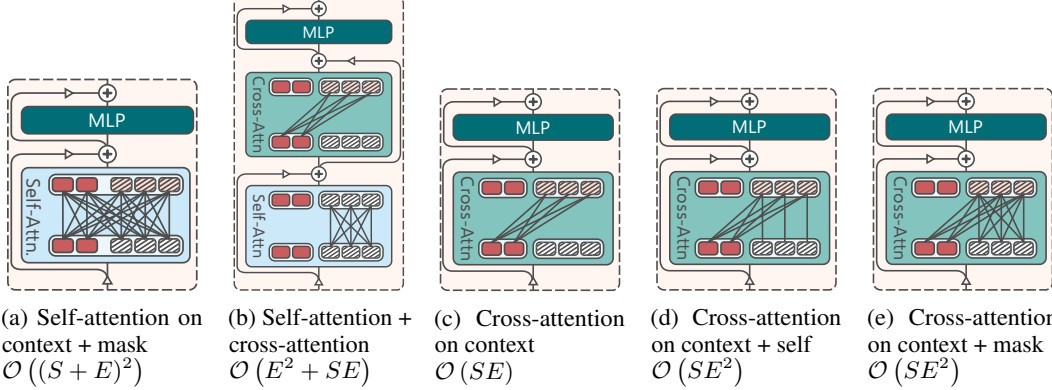

(a) Self-attention on context + mask
$\mathcal{O}\left((S+E)^2\right)$

(b) Self-attention + cross-attention
$\mathcal{O}\left(E^2+SE\right)$

(c) Cross-attention on context
$\mathcal{O}\left(SE\right)$

(d) Cross-attention on context + self
$\mathcal{O}\left(SE^2\right)$

(e) Cross-attention on context + mask
$\mathcal{O}\left(SE^2\right)$

Figure 13: Different transformer block designs for the predictor. The time complexity of the attention computations is included, with $S = |\tilde{\mathcal{M}}|$ and $E = |\mathcal{M}|$ the number of context and masked patches respectively.

Table 14: Ablations on the predictor of **AsymSD-MPM-S**. `RG` indicates a regression loss. `MB` indicates the use of multi-block. Of the attention tokens, `C` is the visual context, `M` the mask queries, `S` the query token itself. Mem. is the total memory usage with a batch size of 128 (with multi-mask set to 8) in GiB; and It/s the throughput in iterations per second. For more details of the design refer to the indicated figures.

| Method | Fig. | MB | Attention Self | Attention Cross | Mem. | It/s | ModelNet40 SVM | ModelNet40 FFt | ScanObjectNN SVM | ScanObjectNN FFt |
|---|---|---|---|---|---|---|---|---|---|---|
| **(MPM-S)** | 13c | × | × | c | 16.2 | 4.80 | **93.31** | **94.04** | **81.06** | **88.58** |
| | 13d | × | × | c+s | 17.5 | 4.65 | 93.03 ↓ 0.28 | 94.04 → 0.0 | 81.06 → 0.0 | 87.96 ↓ 0.62 |
| | 13a | × | c+m | × | 17.9 | 4.32 | 88.17 ↓ **5.14** | 92.87 ↓ **1.17** | 77.00 ↓ **4.06** | 85.98 ↓ **2.60** |
| + Multi-Block | 13b | ✓ | m | c | 15.6 | 3.68 | 93.19 ↓ 0.12 | 93.88 ↓ 0.16 | 81.85 ↑ 0.79 | 87.51 ↓ 1.07 |
| | 13e | ✓ | × | c+m | 15.4 | 4.01 | 93.07 ↓ 0.24 | 94.29 ↑ 0.25 | 80.95 ↓ 0.11 | 88.17 ↓ 0.41 |

