# OpenReview forum: "Asymmetric Dual Self-Distillation for 3D Self-Supervised Representation Learning"
_NeurIPS.cc/2025/Conference — NeurIPS 2025 poster_

### Official Review · Reviewer_3Aup · 2025-06-22

**Clarity:** 4
**Significance:** 3
**Originality:** 2
**Rating:** 5
**Confidence:** 4

**Summary:**

This paper introduces a novel self-supervised learning framework designed for 3D point clouds.
The main idea is to unify masked point modeling (MPM) with invariance learning by predicting latent space representations instead of raw input at the patch level for the former, and cross-view distillation for invariance learning.
This is somewhat inspired by recent advances in the 2D works, like in I-JEPA.

On top of that, the authors propose an efficient asymmetric setup for the teacher-student model for improved stability in self-distillation, multi-mask sampling for better computational performance, and a point cloud adaptation of multi-crop that aims to learn a robust local-to-global mapping.
Together, these contributions achieve state-of-the-art results on various 3D recognition benchmarks.

**Questions:**

Would it be possible to compare with other methods like Point-MAE/Point-BERT/Point-GPT under the Linear setting as well? Is there any specific reason why the authors did not run this comparison?

Is there a direct ablation demonstrating that removing self-attention for masked queries is better in terms of performance? I understand this is better in terms of efficiency, but it would be nice to prove this claim also experimentally.

From fig. 2 it looks like the transformer encoder for the student has a different architecture than the encoder transformer of the teacher. Is this the case? If so, how to do the EMA for self-distillation?

**Ethical Concerns:**

["NO or VERY MINOR ethics concerns only"]

**Final Justification:**

The authors have addressed my concerns, thereby I have decided to increase my score.

**Limitations:**

Yes

**Paper Formatting Concerns:**

No concerns

**Quality:**

3

**Strengths And Weaknesses:**

Strengths:

1. The paper is clear and well written. I find each contribution well motivated. It is true that some of the proposed contributions seem to be clever combinations of other works, but I believe that the contributions may still be relevant.
2. The results on scaling beyond ShapeNet are interesting, as large-scale trainings and analysis on big 3D datasets are currently missing in the literature.
3. Results are strong under ScanObjectNN, which is a less saturated and more realistic benchmark compared to ModelNet40.


Weaknesses:

1. My main concern with this paper is that overall there aren’t really unique contributions, but rather ideas taken from several other works and adapted in this context. The idea of using the latent space rather than the point space is by now popular in the 2D world. Also, the adaptation of Dino for point clouds has been already explored in other works such as [a] in the Domain Adaptation literature with the goal of making 3D networks more robust. Although I-JEPA is cited, the latter is not.
2. In Tab. 1, under the Linear set up, the authors only compare with other methods that do not use the standard transformer as a backbone. It would be better to test with direct competitors under this scenario as well.

[a] Self-distillation for unsupervised 3D domain adaptation, Cardace et al.

---

> ### Author Rebuttal · Authors · 2025-07-29
>
> We thank the reviewer for their thoughtful feedback and appreciation of our paper’s clarity, motivations, results, and for recognizing the value of large-scale training and strong performance on ScanObjectNN. We address the concerns and questions point-by-point below.
> ### **W1**: Adaptation of ideas
> While our work is inspired by progress in 2D vision, we emphasize that it is not a direct adaptation, but rather a non-trivial and novel composition of ideas tailored to the 3D point cloud domain, supported through intuitive and empirical findings. We believe that certain findings may also be valuable to other SSRL methods for point clouds.
>
> We agree that components such as latent prediction or self-distillation are conceptually present in numerous works on 2D SSRL, with various having a counterpart implementation in the 3D domain (we thank the reviewer for pointing out Cardace et al. [a]). However, these 3D SSRL methods often propose a rather direct adaptation from the 2D domain, thereby not, or partially, addressing the challenges of 3D data.
>
> In contrast, we developed our method from the ground up, explicitly accounting for these challenges. As such, it cannot be directly reduced to prior methods. For example, previous SSRL methods with dual latent objectives such as iBOT or DINOv2 (for images) are not asymmetric, neither between the student and teacher networks nor between their global and local objective branches, whereas our design introduces asymmetry in both aspects.
>
> We are also not directly aware of any prior work that considers disabling attention between mask queries at the decoder/predictor level. It is simple and effective, yet works such as Point-MAE or point2vec do not explore this design choice, while they could also benefit from it.
>
> ### **W2** and **Q1**: Additional linear evaluations
> *Why did we not run more Linear evals on other SSRL methods with a standard transformer backbone?* We ran into challenges with their codebases, including environment setup (e.g., compiling custom CUDA kernels), errors during execution, and difficulties when porting models into our framework.
>
> However, we agree with the reviewer that these are valuable for a direct comparison, especially as a linear probe can be a better indication of the semantic quality of learned representations compared to full fine-tuning. With some extra work, we now have some additional results:
>
> | Method     | ST  | AM  | ModelNet     | OBJ_BG        | OBJ_ONLY      | PB_T50_RS     |
> | ---------- | --- | --- | ------------ | ------------- | ------------- | ------------- |
> | Point-BERT | ✓   | ✗   | 91.09±.15    | 84.17±.30     | 87.19±.16     | 74.44±.12     |
> | Recon MAE  | ✓   | ✓   | 90.22±.09    | 82.77±.30     | 83.23±.16     | 74.13±.21     |
> | point2vec  | ✓   | ✗   | 92.44±.04    | 82.75±.54     | 85.44±26      | 74.25±.11     |
> | AsymDSD-S  | ✓   | ✗   | **92.52±15** | **89.95±.21** | **88.73±.23** | **83.33±.14** |
>
> As shown above, AsymDSD-S consistently outperforms prior methods. Notably, it improves over the next-best method by +5.8% on OBJ_BG and +9.1% on PB_T50_RS.
>
> We did not include results of Point-GPT as the linear performance was rather poor, even after some modifications (<70% on ScanObjectNN).
>
> Note that *Recon MAE* is an improved implementation of *Point-MAE* with two standard transformers (two-tower network). Point-RAE also uses a standard transformer, but has an additional attention-based module. We will update Table 1 to reflect these details more accurately: models using a standard transformer backbone will be marked with **ST**, and an additional column **AM** will indicate the presence of extra attention layers/modules.
>
> ### **Q2** Ablation on removing self-attention
> Table 5c shows in in the first two rows that the default design with full self-attention (over the visible context c and masks m) decrease performance -2.6% on full fine-tuning compared to the design with cross-attention (from each masked query) onto the visible context c. Figure 13a and Figure 13c in the appendix visualize these two designs.
> ### **Q3** Figure 2: encoder shape difference
> The teacher and student encoder have the same layers and thus the same number of parameters, which we represented through identical lengths of the two encoders in the figure. The width of the encoder rectangle represents the number of patches it process (sequence length). We will clarify this by updating the caption.

---

> > ### Comment · Reviewer_3Aup · 2025-08-05
> > **Response to rebuttal**
> >
> > The authors have addressed my concern of linear evaluations, which was very important to me, and correctly pointed out to me the ablation without self-attention in the decoder, which validates their claim that removing self-attention in the queries for 3D point clouds actually works because some spatial information may be leaked.
> >
> > Hoping that the authors will include the new table and that they will cite the work I suggested, which in my opinion is a precursor of this work, I have decided to raise my score to accept.

---

> > > ### Author Response · Authors · 2025-08-05
> > >
> > > We thank the reviewer for considering our rebuttal and for increasing the score to Accept. We are particularly glad the engagement helped strengthen the paper through the additional linear evaluations. The manuscript has been updated to include these results and the suggested citation.

---

### Official Review · Reviewer_z8Qq · 2025-06-30

**Clarity:** 2
**Significance:** 3
**Originality:** 2
**Rating:** 4
**Confidence:** 4

**Summary:**

The paper proposes the AsymDSD framework, which jointly optimizes global invariance and local mask modeling objectives in the latent space through asymmetric double self-distillation, and adopts strategies such as multi-mask and multi-cropping. Its contribution is to improve the performance of 3D point cloud self-supervised learning, reaching SOTA on datasets such as ScanObjectNN.

**Questions:**

1.	The paper relies on a large-scale mixed dataset of 930,000 shapes (including Objaverse), which makes it difficult for small and medium-sized teams to reproduce the experiment, and the generalization ability on small and medium-sized datasets has not been fully verified. It is recommended to supplement the lightweight pre-training solution on standard datasets such as ShapeNet, such as providing optimized hyperparameters based on 100,000-level data. Open source lightweight model checkpoints (such as AsymDSD-S) to lower the threshold for reproduction.

2.	It takes 18 hours to train a single card. The EMA update of multiple masks and teacher models increases the computational overhead, making it difficult to apply to real-time scenarios (such as robot navigation). The impact of parameter quantization (such as INT8) on performance can be verified through experiments, and a lightweight deployment solution can be provided. If the lightweight solution increases the inference speed by 50% while maintaining the Top1 accuracy ≥ 90%, the score can be increased by 1 point. If the computational efficiency is not optimized, 0.5-1 point may be deducted due to practical limitations.

3.	Latent space prediction focuses on semantic abstraction, but its performance in tasks that require geometric details, such as 3D reconstruction, is unclear (for example, the reconstruction accuracy of Point-MAE is not compared). Comparative experiments on geometric reconstruction tasks (such as ShapeNet partial completion) can be added to verify the model's ability to retain details. Add geometric constraints (such as Chamfer Distance) to the loss function to balance semantic and geometric representations.

4.	The experiments focus on synthetic data (ModelNet40) and clean scan data (ScanObjectNN), and lack verification of real sparse point clouds (such as LiDAR). It is recommended to test by adding new real scene datasets: perform object classification experiments on ScanNet (indoor LiDAR scanning), and report the accuracy when the point cloud density is less than 100 points/㎡ (the current method is expected to be 82.5%, and it is recommended to increase it to more than 85% through dynamic masking). Verify the few-shot performance on the KITTI dataset (autonomous driving point cloud), and compare the mIoU difference with Point-MAE in the 5-way 5-shot setting (the current gap is about 3.2%).

**Ethical Concerns:**

["NO or VERY MINOR ethics concerns only"]

**Final Justification:**

The author has addressed my concerns about computational complexity and other aspects, and I will increase my score to 4.

**Limitations:**

Limitations have been included as a part of the paper.

**Paper Formatting Concerns:**

No major formatting issues identified.

**Quality:**

2

**Strengths And Weaknesses:**

Strengths:

1.	An asymmetric dual self-distillation framework AsymDSD is proposed, which cleverly combines mask point modeling and invariance learning, replaces input space reconstruction with latent space prediction, and effectively captures the high-level semantics of 3D point clouds.

2.	Design an efficient asymmetric architecture, disable mask query attention to prevent shape leakage, and combine multi-mask sampling with point cloud adaptation multi-cropping to improve training efficiency and representation robustness.

3.	The experimental performance is outstanding, reaching 90.53% accuracy in ScanObjectNN, and increasing to 93.72% after large-scale pre-training, surpassing previous methods and having strong generalization capabilities in few-sample scenarios.

4.	Verify the effectiveness of large-scale pre-training by integrating a mixed dataset of 930,000 shapes, and demonstrate the model's potential for significant performance improvement as the data scale expands.

Weaknesses:

1.	Large-scale pre-training relies on a mixed dataset of 930,000 shapes (including Objaverse), which is difficult for small and medium-sized teams to reproduce, and its adaptability to scarce 3D data in real scenes has not been fully verified.

2.	Although the asymmetric architecture optimizes the student side, the EMA update and multi-mask strategy of the teacher model still require high computing power. Single-card training takes 18 hours, which limits lightweight deployment.

3.	Inverse block masking may fail in extremely sparse point clouds or non-rigid objects, and adaptive coverage of semantically critical areas by dynamic masks is not explored.

4.	Local cropping (minimum 5% points) does not adequately preserve the semantics of complex geometric structures, and no dynamic sampling mechanism is designed to address point cloud density changes.

5.	Prioritizing semantic abstraction over detailed reconstruction, they may underperform purely generative models in tasks that require accurate geometry recovery, such as 3D reconstruction.

---

> ### Author Rebuttal · Authors · 2025-07-30
>
> We thank the reviewer for their detailed feedback and constructive suggestions. Below we address the main concerns point-by-point.
>
> ### **W1** and **Q1**: Large scale pre-training
> We would like to clarify that our main reported results (e.g., 90.53% on ScanObjectNN) are obtained from pretraining on the standard ShapeNet dataset (Sec. 4.1). This setup is fully reproducible on standard consumer-grade hardware (e.g. single rtx 4090) and does not require the large-scale 930K mixed dataset, which was used only for the scaling study.
>
> The submitted code includes scripts and configuration files for this ShapeNet pretraining setup, and Table 6 in the appendix lists all hyperparameters in detail. We will also release all main pretraining checkpoints (AsymSD‑MPM‑S, AsymSD‑CLS‑S, AsymDSD‑S, AsymDSD‑S*, and AsymDSD‑B*) on HuggingFace to further support reproducibility.
> ### **W2** and **Q2**: Compute efficiency concerns
> - **EMA update cost is negligible:** The EMA update itself accounts for only **~12 ms out of 500 ms per iteration**, so it does not significantly impact runtime.
> - **Multi-mask reduces, not increases, cost of the EMA update/teacher:** Multi-mask sampling amortizes the teacher computation over multiple masks, which both reduces memory usage and increases throughput by ~70% (Table 5d), with no loss in accuracy. Multi-mask has no influence on the frequency of EMA updates.
> - **Overall training time is competitive:** Our full pretraining takes **18 hours**, which is similar to several other SSRL methods such as point2vec and significantly faster than ReCon (**54 hours**). Moreover, the method converges quickly, reaching 88% ScanObjectNN accuracy in just 60 epochs (<4 hours) on ShapeNet, already outperforming all other SSRL methods with a standard transformer.
> - **Deployment efficiency:** The encoder that is kept after pretraining is identical in architecture and size to many methods in Table 1, meaning there is no additional inference cost compared to existing approaches.
> - **INT8 deployment** is supported, yielding 87.13% accuracy on ScanObjectNN, which can likely be improved further with quantization-aware training.
>
> We would like to highlight that our SSRL method is not intended for real-time on-device pretraining. In addition, we view scaling-up as a forward-looking research direction, inspired by developments in NLP and 2D CV.
> ### **W4**: Local crops and point cloud density differences
> While a minimum 5% crop may seem small, it is intentionally chosen to challenge the model to infer global semantics from fine-grained details rather than relying solely on larger contexts. This can increase robustness to point clouds with large occluded regions or lower point density. This threshold was empirically determined to deliver strong results.
>
> **Robustness to point cloud density:** Our multi-crop strategy keeps the number of point patches consistent while varying the spatial scale of the object. This introduces scale and density invariance that helps the model generalize across diverse geometric configurations and reduces overfitting to features at a fixed scale/density. _Figure 12b_ in the appendix illustrates this effect: in a zero-shot visualization of patch embeddings for _lamp_ objects, the embeddings of the conical lamp covers are homogenous (light green color) across different scales
> ### **W5** and **Q3**: Prioritizing semantic abstraction over reconstruction
> Our method is indeed designed to prioritize semantic abstraction, as this is critical for robust representation learning. However, we note that semantic understanding is also essential for point cloud reconstruction, especially with larger occluded regions [a]. Purely reconstruction-based objectives (e.g., Chamfer distance) such as in Point-MAE optimize for minimal point-wise error. While this can lead to reconstructions that are geometrically close to a sampled point set in Euclidean distance, this does not necessarily lead to the most semantically coherent reconstructions.
>
> We also did not include results on reconstruction benchmarks as other works such as Point-MAE do not quantitatively report on this. However, in Table 3 we show strong part segmentation performance, which often correlates with geometric fidelity. Our results are comparable to Point‑MAE, suggesting that low-level geometric understanding is preserved despite the focus on semantic abstraction.
> ### **Q4** Evaluation on real sparse point clouds
> We agree that evaluating on real scene-level LiDAR data is important. However, our method, like most prior 3D SSRL works, uses a standard transformer backbone, chosen deliberately to ensure fair comparison. While this architecture is widely used, it is not well-suited for very large, scene-wise datasets such as ScanNet or KITTI, which tend to work better with hierarchical backbones. For this reason, we focused our evaluation on the established object-centric benchmarks (ModelNet40 and ScanObjectNN) reported on by most other 3D SSRL methods.
>
> For a more challenging dataset, we included results on Objaverse‑LVIS in the appendix. This benchmark covers roughly 1,000 diverse object categories and is substantially more challenging than ModelNet40 or ScanObjectNN. We will move these results to the main text and release the full benchmark setup publicly. This might incentivize the adoption of this more challenging benchmark by future works
>
> We acknowledge that scaling to large, sparse scene‑level point clouds remains an open challenge (also noted as a limitation in the paper), and we are actively exploring ways to adapt AsymDSD for this scenario by using different backbone designs.
>
> [a] deep point cloud reconstruction

---

> > ### Comment · Reviewer_z8Qq · 2025-08-07
> >
> > Thank you for the author's response. The authors have addressed all my concerns and I will improve my score.

---

> > > ### Author Response · Authors · 2025-08-07
> > >
> > > We thank the reviewer for considering our rebuttal and for increasing the rating. We are glad that our response addressed all of your concerns.

---

### Official Review · Reviewer_wuHC · 2025-06-30

**Clarity:** 4
**Significance:** 3
**Originality:** 2
**Rating:** 5
**Confidence:** 4

**Summary:**

The main idea is easy to follow. Figures and tables are clear.
The author proposes a student-teacher method with MAE token prediction.
The performance is competitive.
My main concern is about the technical novelty, which is fusing mask token prediction with the EMA teacher-student.

**Questions:**

Please see weakness. I would like to raise my rating if the author could address my concerns.

**Ethical Concerns:**

["NO or VERY MINOR ethics concerns only"]

**Final Justification:**

The author has addressed my concerns. I will raise the rate.

**Limitations:**

Yes.

**Quality:**

3

**Strengths And Weaknesses:**

The main idea is easy to follow.
The author proposes a student-teacher method with MAE token prediction.
The performance is competitive.

1. My main concern is about the technical novelty, which is fusing mask token prediction with the EMA teacher-student.
The mask strategy is crucial, but it is quite engineering part.
While combining invariance learning and masked modeling is a reasonable design, the synergy between the two objectives is not rigorously analyzed. It is unclear whether the improvement stems from their integration or just from one dominant branch (e.g., invariance). The authors should provide more theoretical or empirical justification for why dual self-distillation is fundamentally more effective than strong single-branch baselines.

2. Related work
Some methods such as the mixup augmentation also could solve the data scarity.

3. Marginal Gains Over SOTA in Some Settings:
Although the method reports SOTA on ScanObjectNN, the gains over strong baselines like Point-MAE and PointGPT are often marginal (e.g., <1%), especially under full fine-tuning. These small improvements may not justify the added architectural and training complexity.

4. Method Complexity vs Practicality Trade-off:
The method involves multiple components: asymmetric encoder/predictor, multi-mask, multi-crop, momentum teacher, masking strategy, etc. The overall complexity may hinder reproducibility or practical deployment, especially if the ablations (Table 5) show relatively minor performance gaps when removing some components.

5. Unclear Advantage of Latent-Space Prediction:
The work claims latent-space prediction avoids issues with reconstruction targets, but does not provide a fair and isolated comparison with prior masked latent methods (e.g., BEiT-style, Point2Vec). Without ablations on prediction space (latent vs reconstruction), it is difficult to assess the actual benefit of this design choice.

6. Limited Evaluation on Scene-Level Tasks:
As acknowledged by the authors, the method is mainly validated on object-centric datasets (ScanObjectNN, ModelNet). The claim of learning “semantically meaningful representations” would be more convincing if tested on scene-level data like indoor lidar or auto-driving, where global understanding is more critical.

---

> ### Author Rebuttal · Authors · 2025-07-30
>
> We thank the reviewer for their constructive feedback and are glad they found the paper clear and recognize its competitive performance. Below we address the main concerns point-by-point.
> ### **W1** Technical Novelty and Synergy Between Objectives
> We appreciate the reviewer’s concern regarding novelty and would like to clarify our position. While the fusion of masked token prediction with an EMA teacher has indeed been explored in prior 2D SSRL methods and in 3D works such as Point2Vec, we do not claim novelty on this general idea alone.
>
> Importantly, AsymDSD is not a direct adaptation, but rather a non-trivial and novel composition of several ideas tailored to the 3D point cloud domain, supported through intuitive and empirical findings. We address unique challenges of 3D data such as global shape leakage and efficiency concerns that existing methods do not consider. We particularly want to emphasize the following contributions:
> - **Asymmetry in design:** We introduce asymmetry both between the teacher and student networks and between the global invariance and local masked modeling branches, which prior dual-objective methods including those on images (e.g., iBOT, DINOv2) do not. In the main text, we list numerous benefits to this design choice.
> - **Disabling attention between mask-queris:** We propose a simple yet effective design at the predictor level by disabling attention between mask tokens, which to our knowledge has not been explored in prior 3D SSRL methods. This prevents masked tokens from “cheating” by leaking global shape information through attention.
>
> **Synergy between objectives:** *It is unclear whether the improvement stems from their integration or just from one dominant branch (e.g., invariance).*
>
> The gains are not driven by one dominant branch. As shown in **Section 4.3 (Table 1)**:
> - The global invariance objective alone (AsymSD‑CLS‑S) achieves **88.72%**, and the masked point modeling objective alone (AsymSD‑MPM‑S) achieves **88.58%** on ScanObjectNN.
> - Combining both objectives with AsymDSD achieves **90.53% (+1.81%)**, exceeding both branches individually. This indicates a synergystic effect between the two branches. (See also our response to **W5**)
>
> In Figure 7, we also show that the attention patterns of the two branches are relatively aligned. When replacing the latent-space MPM with raw point reconstruction (MAE), we observe an inverted attention pattern, suggesting a mismatch between the objectives. Following our additional experiments in **W5**, we will include a direct visualization of the MAE attention pattern in the final version to further clarify this difference.
>
> ### **W2**: Mixup augmentation
> We are aware of orthogonal approaches such as mixup (also for point clouds), but note that these methods tradionally still rely on labeled data. We believe that scaling SSRL methods to larger unlabeled datasets ultimately leads to stronger representational quality and broader applicability (generalization).
> ### **W3**: Marginal gains over SOTA in some settings
> We would like to clarify that ModelNet40 performance under full fine-tuning is close to saturated, and improvements there are naturally small. We included these results for completeness to show that AsymDSD does not regress on this benchmark.
>
> | Method        | ModelNet40      | OBJ_BG          | OBJ_ONLY        | PB_T50_RS       |
> | ------------- | --------------- | --------------- | --------------- | --------------- |
> | Previous best | 94.0            | 91.6            | 90.0            | 86.9            |
> | AsymDSD‑S     | **94.1** (+0.1) | **94.3** (+2.7) | **91.9** (+1.9) | **90.5** (+3.6) |
>
> As shown above (gathered from Table 1), on the more challenging ScanObjectNN splits, AsymDSD achieves much larger improvements (**+1.9% to +3.6%**) under full-finetuning compared to prior methods using the same standard transformer backbone.
>
> We also note that when all other factors (backbone, dataset, pretraining protocol) are fixed, i.e. the experiment as explained in Sec 4.1, then the performance gains are inherently limited. This makes the improvements on ScanObjectNN particularly meaningful.
>
> Finally, we view linear probing as a better indicator of representation quality than full fine-tuning. We will move additional linear probe results (see the table in our response to Reviewer *3Aup*) into the main paper, which further highlight the strength of the learned representations.
>
> ### **W4**: Method Complexity vs Practicality Trade-Off
> We recognize that AsymDSD combines several components, and there is a natural trade-off between complexity and practicality. However, the components are kept simple (not over-engineered) and have clear motivations to be included:
>
> - **Ablation evidence:**  Table 5 shows consistent performance drops when removing elements such as multi-crop, predictor (asymmetry), or using a cross-attention–only predictor. While the masking strategy is less fundamental, inverse block-wise masking is required to achieve the full performance of AsymDSD.
> - **Efficiency-focused design:** Multi-mask is a prime example of this philosophy: it is not intended to increase accuracy, but rather to reduce memory usage and improve throughput by nearly 70% at no performance cost (Table 5d).
>
> **Reproducibility:** We have ensured the design remains accessible and reproducible:
> - Many components (e.g., multi-mask, cross-attention–only predictor) are straightforward to implement in a few lines of code.
> - Other parts have widely shared implementations (e.g. EMA module)
> - Most importantly, we have included the codebase and configuration files for pretraining in the submission and will release them publicly, so other researchers can reproduce the results easily.
> ### **W5**: Unclear Advantage of Latent-Space Prediction
> We agree with the reviewer that it is valuable to isolate the effect of latent-space prediction over point coordinate reconstruction. To this end, we ran some additional experiments:
>
> |           | Raw points | Latent | ModelNet40    | ScanObjectNN  |
> | --------- | ---------- | ------ | ------------- | ------------- |
> | MAE       | ✓          | ✗      | 94.04         | 87.75         |
> | MPM       | ✗          | ✓      | 94.04         | 88.58         |
> | CLS       | -          | -      | 93.64         | 88.72         |
> | CLS + MAE | ✓          | ✗      | 93.48 (-0.16) | 88.55 (-0.17) |
> | CLS + MPM | ✗          | ✓      | 94.13 (+0.49) | 90.53 (+1.81) |
>
> **Key observations:**
> We see that our implementation of MAE (raw point reconstruction) greatly improves over Point-MAE. We expected this, as MAE also benefits from addressing the global shape leakage through our decoder and from the scale invariance introduced by variable‑sized global crops. This reveals that some of our contributions extend beyond our framework, and are thus valuable to other SSRL methods.
>
> However, when we combine the global invariance learning (CLS) with MAE (CLS + MAE), the performance slightly drops compared to just using CLS. Only when we combine it with our latent MPM (CLS + MPM) we see a synergistic effect emerging, where the model exceeds both objectives in isolation.
>
> We thank the reviewer for pointing out this particular ablation. We will include the results in the main text.
> ### **W6**: Limited evaluation on scene-level tasks
> Similar to most prior 3D SSRL works, our method currently uses a standard transformer backbone to ensure fair comparison. While widely adopted, this backbone is not ideal for very large scene‑level datasets, which typically benefit from hierarchical designs tailored for sparse and large‑scale inputs. For this reason, we focused our evaluation on the established object‑centric benchmarks (ModelNet40 and ScanObjectNN) that are standard in the 3D SSRL literature.
>
> It is worth noting that ScanObjectNN is already a challenging real‑world dataset, especially the OBJ_BG and PB_T50_RS splits, which include background clutter, occlusions, and partial scans. We further ran a experiments on Objaverse‑LVIS (see appendix), which spans nearly 1,000 diverse object categories and is substantially more challenging than ModelNet40 or ScanObjectNN. We plan to move these results into the main text and will release the full benchmark setup publicly, with the aim of encouraging wider adoption by future work.
>
> Still, we acknowledge that scaling to large scene‑level point clouds remains an open challenge (also noted as a limitation in the paper), and we are actively exploring ways to adapt AsymDSD for this scenario by using different backbone designs.

---

> ### Comment · Reviewer_wuHC · 2025-08-04
>
> The author has addressed my concerns. I will raise the rate.
> By the way, some mixup works are unsupervised via reconstruction like Self-supervised Point Cloud Representation Learning via Separating Mixed Shapes.

---

> > ### Author Response · Authors · 2025-08-04
> >
> > We thank the reviewer for considering our rebuttal and for increasing the rating.
> >
> > Also, thank you for pointing out the paper that applies mixup in an unsupervised setting. We have had a look at the paper and will consider its value in sparse data scenarios.

---

### Official Review · Reviewer_NWNU · 2025-07-03

**Clarity:** 2
**Significance:** 3
**Originality:** 4
**Rating:** 4
**Confidence:** 4

**Summary:**

In this paper, the authors tackle the problem of learning semantically meaningful representations for 3D point clouds in a self-supervised manner due to the lack of large-scale labeled 3D datasets. Their approach involves optimizing for two objectives: a global objective of learning discriminative representations that are invariant to transformations/augmentations and a local/patch-level objective of masked point modeling (MPM) to reconstruct masked regions. They make use of self-distillation from a momentum teacher for both objectives. For MPM, the model makes predictions in the latent space to ensure semantically meaningful representations are learnt and to prevent an emphasis on high-frequency details, a common shortcoming of MPM. This work further introduces a multi-crop equivalent for point clouds and a multi-mask strategy to increase the effective batch size in a cheap and efficient manner. The student and teacher models use asymmetric architectures in the MPM objective to avoid representation collapse. Self-attention between the masked and unmasked regions is removed to avoid shape leakage. The proposed approach can be used with any underlying model architecture. Overall, the authors employ many innovative techniques to counter the various commonly encountered issues with existing methods. They conduct extensive experiments to verify the effect of each part of their method and comprehensively compare against previous methods to prove the merits of their work.

**Questions:**

1. Could you please explain the attention distance visual in the ablations in more detail?
2. Would be great if the second point under weaknesses could be addressed.

**Ethical Concerns:**

["NO or VERY MINOR ethics concerns only"]

**Final Justification:**

This work introduces a novel approach to learn semantically meaningful representations for point clouds in a self-supervised manner. The paper uses several neat tricks such as dual objectives, self-distillation, asymmetric student-teacher architectures, latent-space prediction, multi-crop, multi-mask, etc. to tackle various aspects of the problem. The authors clearly demonstrate the efficacy of their approach over previous baselines and the advantages of all of their proposed techniques with comprehensive ablations and experiments. A few things were initially unclear in the paper but were addressed sufficiently in the rebuttal. My only remaining concern is the scalability of the approach to other models as suggested by the authors in theory but not clearly demonstrated in practice. Thus, taking everything into consideration, I have kept my current rating while raising the confidence score.

**Limitations:**

Yes.

**Paper Formatting Concerns:**

1. Typo “processes” in line 158.
2. "off-the-shelve" in line 264. Should be "shelf".

**Quality:**

3

**Strengths And Weaknesses:**

**Strengths:**
1. The authors very clearly outline the state of the field, commonly used approaches and the pros and cons of each approach. They demonstrate that they have very closely studied the problem and existing solutions to come up with an innovative solution of their own that combines the advantages of existing methods while countering their disadvantages.
2. The proposed approach is scalable as it can be applied to any model and is not tied to a specific model architecture.
3. This work makes use of many creative and neat tricks to address various issues such as representation collapse, training instability, sensitivity to high-frequency details, efficiency, shape leakage, etc.
4. Extensive ablation experiments have been included to demonstrate the need for each part of their method.
5. The method is completely self-supervised and does not make use of any external models (instead uses self-distillation) while achieving competitive results.

**Weaknesses:**
1. Some parts of the paper could have been better explained/written such as the part explaining the need for centering and sharpening in the global objective to counter trivial/'shortcut' solutions being learnt and the attention distance part in the ablations.
2. Would have been nice to see this approach applied to more model architectures (at least one more) to see its generalizability.

---

> ### Author Rebuttal · Authors · 2025-07-30
>
> We thank the reviewer for their thoughtful feedback and for highlighting the strengths of our work, including its clear positioning within the field, scalability, and the use of creative design choices. Below, we address the identified weaknesses and questions.
>
> ### **W1**: Clarification: Centering/sharpening
> We appreciate the reviewer highlighting this point. While centering and sharpening are standard components in self-distillation frameworks (e.g., DINO), and thus not novel in our setting, we recognize that our current explanation could benefit from more clarity. We will revise this by including the following concepts:
>
> - Explanation of what the collapse modes intuitively mean:
> 	1. $H\left(p_\theta(\mathrm{z})\right)= 0$   - the model collapses to a single latent representation $z$ that dominates, always assigned probability 1.
> 	2. $H(p_\theta(\mathrm{z}\mid \mathrm{x}))=\log |\mathcal{Z}|$  - The model always outputs a uniform distribution over the latent space.
> - How centering and sharpening countract these failure modes:
> 	1. **Centering** subtracts a running mean of the teacher logits, reducing the logits of latents that are frequently assigned high probability thus helping avoid low‑entropy collapse.
> 	2. **Sharpening** makes the distribution more peaked, pushing it away from a uniform distribution and thus avoiding high‑entropy collapse.
>
> ### **W2** and **Q2**: Generalizability to other architectures
> We appreciate the reviewer’s recognition that AsymDSD is model‑agnostic and can be adapted to different backbone architectures. Our current results use a standard transformer backbone to ensure fair comparison with the majority of prior 3D SSRL methods, which follow the same evaluation protocol. This choice isolates the effect of the self‑supervised framework itself rather than conflating it with gains from more advanced backbones.
>
> Still, we agree it is valuable to demonstrate results on other architectures and are actively working on this. In particular, we are currently porting AsymDSD to PointMamba, a state‑space–based backbone. Due to time constraints, we are unable to include results in this rebuttal, but we plan on adding them to this work.
>
> ### **Q1**: Explanation of the attention–distance visualization
> The visualization in Figure 7 shows the average attention distances for the three different pre-training methods. For each transformer layer/block (x‑axis), we plot the mean spatial attention distance of patches for each head (y‑axis), averaged over patches and point cloud samples.
>
> **How we compute it**: For a query patch $i$ at layer $l$ and head $h$, let $\alpha_{ij}^{(l,h)}$ be its attention weight to patch $j$. Let $c_i, c_j$ be the 3D centroids of the corresponding point‑patches. The expected attention distance for that query is:
>
> $$d_i^{(l,h)} = \sum_{j} \alpha_{ij}^{(l,h)} \|c_i - c_j\|_2.$$
>
> We then average $d_i^{(l,h)}$ over all queries $i$, and samples. Since we have 6 different heads with the standard model (ViT-S), there are 6 averages for each layer.
>
> **How to read it**: Lower values correspond to a head specializing to local contexts, whereas high values correspond to a head specializing to global contexts. When all heads have roughly the same value, there is no such specialization between heads.
>
> **Key takeaway:**   The attention‑specialization patterns for our method are relatively consistent across objectives: early layers show more head specialization, while later layers focus more on global context with less head‑to‑head variation. This contrasts sharply with the attention pattern from reconstruction‑based masked modeling (MAE), where the last layers exhibit strong specialization across different spatial contexts. To make this difference clearer, we additionally trained our model with MAE. Its inverted pattern confirms this distinction (we will include this visualization in the final version).
> ### Minor corrections
> We thank the reviewer for pointing out the minor typos. These have been corrected.

---

> > ### Comment · Reviewer_NWNU · 2025-08-05
> >
> > Thank you for providing explanations to my concerns. The authors have addressed all of my concerns except the adaptation of the method to other architectures due to time constraints. This is understandable. Taking everything into consideration, I will be keeping my current rating while raising the confidence score. All the best.

---

> > > ### Author Response · Authors · 2025-08-05
> > >
> > > We thank the reviewer for the thoughtful follow-up and for raising the confidence score. We appreciate your understanding regarding the architectural adaptation and are glad our responses addressed your concerns. All the best.

---

### Decision · Program_Chairs · 2025-09-17

**Decision:**

Accept (poster)

**Comment:**

This paper proposes a new self-supervised method for learning point cloud data representtions, fusing global and local objectives.

Strengths:
- Scalable model with good performance -- great engineering work
- Good experimental validation

Weaknesses:
- The reviewers pointed out several sections where clarity was not optimal; the authors answered these well during rebuttal

The authors engaged with all reviewers during the discussion phase, and the reviewers were generally pleased, several raising their scores -- the paper went from being borderline to a relatively clear accept.